# Enhancing Remaining Useful Life Prediction with Ensemble Multi-Term Fourier Graph Neural Networks

**Ya Song**                                                                      *y.song@tue.nl*
*Industrial Engineering and Innovation Sciences*
*Eindhoven University of Technology*

**Laurens Bliek**                                                                *l.bliek@tue.nl*
*Industrial Engineering and Innovation Sciences*
*Eindhoven University of Technology*

**Yaoxin Wu**                                                                    *y.wu2@tue.nl*
*Industrial Engineering and Innovation Sciences*
*Eindhoven University of Technology*

**Yingqian Zhang**                                                               *yqzhang@tue.nl*
*Industrial Engineering and Innovation Sciences*
*Eindhoven University of Technology*

**Reviewed on OpenReview:** *https://openreview.net/forum?id=tzFjcVqmxw*

## Abstract

Remaining useful life (RUL) prediction is crucial in predictive maintenance. Recently, deep learning forecasting methods, especially Spatio-Temporal Graph Neural Networks (ST-GNNs), have achieved remarkable performance in RUL prediction. Most existing ST-GNNs require searching for the graph structure before utilizing GNNs to learn spatial graph representation, and they necessitate a temporal model such as LSTM to leverage the temporal dependencies in a fixed lookback window. However, such an approach has several limitations. Firstly, it demands substantial computational resources to learn graph structures for the time series data. Secondly, independently learning spatial and temporal information disregards their inherent correlation, and thirdly, capturing information within a fixed lookback window ignores long-term dependencies across the entire time series. To mitigate the issues above, instead of treating the data within the lookback window as a sequence of graphs in ST-GNN methods, we regard it as a complete graph and employ a Fourier Graph Neural Network (FGN) to learn the spatiotemporal information within this graph in the frequency space. Additionally, we create training and test graphs with varying sizes of lookback windows, enabling the model to learn both short-term and long-term dependencies and provide multiple predictions for ensemble averaging. We also consider scenarios where sensor signals exhibit multiple operation conditions and design a sequence decomposition plugin to denoise input signals, aiming to enhance the performance of FGN. We evaluate the proposed model on two benchmark datasets, demonstrating its superior performance on the RUL prediction task compared to state-of-the-art approaches.

## 1 Introduction

The widespread adoption of Cyber-Physical Systems (CPS) and the Internet of Things (IoT) has enabled organizations to leverage predictive maintenance techniques more effectively. This has resulted in extended equipment operational lifecycles, prevented unscheduled downtime, and decreased energy consumption. Predictive maintenance is a proactive approach that estimates the equipment's Remaining Useful Life (RUL),

i.e., forecasts the time point at which equipment may fail or become ineffective in future usage. It then develops appropriate maintenance plans and procedures to ensure the equipment's reliability and continuous operation. Among these, forecasting RUL is considered the most significant and valuable task (Zhou et al., 2021b). RUL prediction typically depends on historical operational data and condition monitoring information, including sensor data, operation records, maintenance histories, and so on. Common prediction approaches include those based on physical models, data-driven methods, and hybrid methods (Ferreira & Gonçalves, 2022). These models examine equipment's operational status and health to predict probable failures. With the advancements in sensing technology and data analytics, data-driven strategies, especially deep learning-based RUL prediction, are emerging as a significant research and application area in engineering.

When utilizing deep learning techniques for RUL prediction, we tackle the problem as a multivariate time series regression task. Generally, we apply the sliding time window approach to generate samples of time series data. The basic idea is to segment raw time series data by sliding a fixed-length time window, taking the data within each time window as a sample, and then using these samples for machine learning model training and prediction. Most current deep learning models for estimating RUL primarily utilize Convolutional Neural Networks (CNNs) (Yang et al., 2019; Ren et al., 2020; Xu et al., 2022), Long Short-Term Memory (LSTMs) (Shi & Chehade, 2021; Wu et al., 2021b), and Transformers (Li et al., 2022; Zhang et al., 2022b; Jiang et al., 2023). These methods have effectively captured the temporal dependencies in time series data. However, they are limited as they do not consider the potential interactions between sensor signals, hindering prediction models' effectiveness. To address this issue, researchers have begun to employ Spatio-Temporal Graph Neural Networks (ST-GNNs) (Jin et al., 2023). This approach involves creating a graph for sensor signal data at each time point, leveraging GNNs to capture spatial information in the graph, and then using sequence models such as LSTMs to encapsulate the temporal information of a sequence of graph embeddings.

Although the experimental results in (Kong et al., 2022; Wang et al., 2023a; 2024) show ST-GNNs can outperform traditional sequence models in RUL prediction tasks, the current ST-GNNs models still have the following four main drawbacks: (1) *Need to learn the graph structure.* Unlike the natural graph structure inherent in problems like traffic flow prediction in road networks, there is no explicit graph structure among sensor signals in RUL prediction. Therefore, it often necessitates the use of graph structure learning methods, demanding substantial computational resources. (2) *Modeling spatial and temporal information separately.* The conventional ST-GNNs separately employ GNN to capture spatial information and LSTM to capture temporal information. This technique fails to consider the possible spatiotemporal inter-dependencies present in sensor signals. (3) *Fixed and short-term dependency modelling.* Traditional models employ a single fixed lookback window to generate samples, often resulting in a window size too small for longer time series data. Consequently, the model struggles to capture long-term dependencies across the entire time series. (4) *Ignore operation condition information.* Equipment may operate under various operation conditions, and analyzing historical operational records can facilitate the learning of potential degradation trends. Existing models solely focus on modeling sensor signals, disregarding operation condition information.

To tackle the above problems, we propose an RUL prediction model named Ensemble Multi-Term Fourier Graph Neural Networks (MT-FGNE). The characteristics and advantages of this model are as follows:

- We adopt a novel approach to time series processing. We no longer view a sample as a sequence of graphs like ST-GNNs; instead, we consider samples as complete graphs. After converting it to the frequency domain using the Discrete Fourier Transform (DFT), we utilize Fourier Graph Neural Networks (FGN) to capture the degradation trends. This approach avoids separately modeling spatial and temporal information, enabling the learning of potential spatiotemporal interdependencies within sensor signal data.

- We propose a multi-term learning framework to address the issue of traditional models' inadequate learning of long-term dependencies. In this framework, we generate training and test graphs with variable lookback window sizes, allowing the model to learn both short-term and long-term dependencies while providing multiple predictions for ensemble averaging.

- We consider the specificity of degradation under multiple operation conditions by incorporating historical operational recording data into the modeling process. After decomposing and interpolating the original signals, we input them into the model and average the prediction results.

- We evaluate our MT-FGNE model on two widely studied benchmarks and achieve competitive performance against state-of-the-art ST-GNN methods.

## 2 Related Work

### 2.1 Deep learning models for RUL prediction

Due to their ability to handle complex nonlinear relationships and perform end-to-end learning, deep learning models have been extensively applied to RUL prediction tasks over the past decade. CNNs excel at extracting local temporal patterns from time series data, while LSTMs specialize in capturing long-term dependencies through their gating mechanisms. The early models consisted mainly of CNNs (Yang et al., 2019; Ren et al., 2020; Xu et al., 2022) and LSTMs (Da Costa et al., 2019; Shi & Chehade, 2021; Wu et al., 2021b). Hybrid models combining both have also been widely utilized, leveraging their capability to extract both spatial and temporal features (Zraibi et al., 2021; Ren et al., 2020). Transformer architectures, leveraging their self-attention mechanisms, can model relationships between all positions in the input sequence simultaneously, enabling more effective capture of both long-term dependencies and global patterns in time series data. Researchers have also applied them to RUL prediction and made various improvements to the attention mechanism (Li et al., 2022; Zhang et al., 2022b; Jiang et al., 2023).

The emergence of GNNs brought new perspectives to spatial information modeling in RUL prediction. Researchers proposed to represent sensor signals as graphs, where each time step corresponds to a graph structure, enabling effective spatial feature learning through various GNN architectures. The main GNN architectures include Graph Convolutional Network (GCN) (Wang et al., 2021; 2023a), Graph Attention Network (GAT) (Zhang et al., 2022b; Kong et al., 2022), and custom Message Passing Neural Networks (MPNNs) (Wang et al., 2023b). In terms of graph construction, one study generates graphs based on domain knowledge (Kong et al., 2022), while another calculates adjacency matrices based on Pearson Correlation Coefficients among sensors (Wang et al., 2021). Recent research prefers to apply graph structure learning approaches. Chen & Zeng (2023) construct the graph structure by computing the cosine similarity between the embedding vectors outputted by GAT. In (Wang et al., 2023a), a dynamic graph learning module is proposed to capture the dynamic relationships between sensor data and generate multi-scale structural insights by dividing data into segments within a lookback window.

In summary, existing ST-GNN methods commonly view samples as sequences of graphs, often requiring graph construction and learning on the original data in the time domain. The FGN method employed in this paper differs significantly in treating samples as complete graphs, eliminating the need to learn graph structures, and transforming the samples into the frequency domain through DFT.

### 2.2 Multi-term learning

The closest idea to our proposed multi-term learning is multi-scale learning. Multi-scale learning is commonly employed in a variety of domains and tasks, including computer vision and time series data analysis. By employing multi-scale learning, CNNs can extract features from receptive fields of different sizes simultaneously, enabling a more comprehensive capture of information in images, including both local details and global structures (Cai et al., 2016). CrossViT (Chen et al., 2021a) divides input images into multiple patches of different sizes and employs a multi-scale feature fusion mechanism to integrate feature representations from different scales. As time series data typically contains patterns and trends at multiple time scales, adopting multi-scale learning in time series analysis helps models gain a more comprehensive understanding and capture structural information within the data (Cui et al., 2016). A classical approach is to apply multi-scale convolution, which generates feature maps at different scales to capture information along the time axis (Chen & Shi, 2021; Chen et al., 2021b). Chen et al. (2023) apply a multi-scale pyramid network to preserve the various temporal dependencies. The model's input remains at equal scales in the aforemen-

tioned multi-scale methods. In contrast, our multi-term learning approach generates inputs of varying scales by employing different lookback window sizes. Some samples contain long-term dependency information, while others only contain short-term information. In the following sections, we will go into the details of the proposed multi-term learning framework.

# 3  Method

The overview of MT-FGNE is shown in Figure 1, which consists of two main components and one plugin. The first component is FGN, which is an individual model to learn spatial and temporal dependencies. The other component is a multi-term ensemble learning framework, which constructs samples at different scales to enable the model to abstract both short-term and long-term dependencies. Considering that some sensor signals are generated when the equipment operates under various operation conditions, we designed a time series decomposition plugin to enhance the model's performance for such inputs.

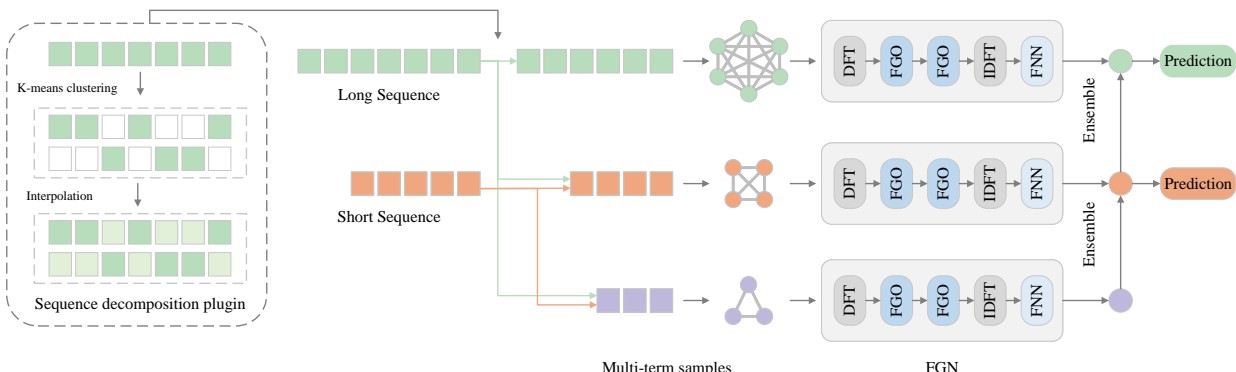

Figure 1: The overall framework of the proposed MT-FGNE. In MT-FGNE, a multi-term ensemble learning strategy is applied to construct samples at different scales to enable the model to abstract both short-term and long-term dependencies, and multiple FGNs are used to learn spatial and temporal dependencies within multi-term samples. Besides, a sequence decomposition plugin is designed to tackle sensor signals recorded under different operation conditions.

## 3.1  Preliminaries and motivations

Given multiple condition monitoring time series data $\left\{X^{(1)}, X^{(2)}, ..., X^{(M)}\right\}$, where $X^{(i)} = \left[\mathbf{x}_1^{(i)}, ..., \mathbf{x}_t^{(i)}, ..., \mathbf{x}_{L_i}^{(i)}\right] \in \mathbb{R}^{L_i \times N}$ represents the $i$-th time series with length $L_i$ and feature dimension $N$, $\mathbf{x}_t^{(i)} \in \mathbb{R}^N$ represents the value of $N$ features at timestamp $t$. We convert the raw time series into samples by applying the sliding time window approach, with the lookback window size $T$, each containing $T$ observations at one time step as input features, and the corresponding output label $Y_t^{(i)}$. $X_t^{(i)} = \left[\mathbf{x}_{t-T+1}^{(i)}, \mathbf{x}_{t-T+2}^{(i)}, ..., \mathbf{x}_t^{(i)}\right] \in \mathbb{R}^{T \times N}$ denotes the input features of one sample at timestamps $t$. The RUL prediction task involves predicting the label $Y_t^{(i)}$ based on the input features $X_t^{(i)}$. When employing traditional sequential models to abstract the temporal information, the prediction process can be formulated by:

$$\hat{Y}_t^{(i)} := F_{\theta_t}(X_t^{(i)}) = F_{\theta_t}\left(\left[\mathbf{x}_{t-T+1}^{(i)}, \mathbf{x}_{t-T+2}^{(i)}, ..., \mathbf{x}_t^{(i)}\right]\right), \tag{1}$$

where $\hat{Y}_t^{(i)}$ are the forecasts matching the ground truth $Y_t^{(i)}$. The forecasting function is denoted as $F_{\theta_t}$ parameterized by $\theta_t$. When using the ST-GNNs method, we first design the graphs or apply graph structure learning approaches to transform $\mathbf{x}_t^{(i)}$ into $\mathbf{g}_t^{(i)}$ at each timestep $t$, and then the RUL prediction can be

expressed as the following formulation:

$$\hat{Y}_t^{(i)} := F_{\theta_t, \theta_g}(X_t^{(i)}) = F_{\theta_t, \theta_g}\left(\left[\mathbf{g}_{t-T+1}^{(i)}, \mathbf{g}_{t-T+2}^{(i)}, ..., \mathbf{g}_t^{(i)}\right]\right), \tag{2}$$

where the forecasting function is denoted as $F_{\theta_t, \theta_g}$ parameterized by $\theta_t$ and $\theta_g$, indicating ST-GNNs separately model spatial and temporal dependencies.

## 3.2  Fourier Graph Neural Networks

A recent study (Yi et al., 2024) introduces FGN to address the oversight of potential spatiotemporal inter-dependencies that arise from modeling spatial and temporal dependencies separately in ST-GNNs. FGN aims to enhance learning efficiency by learning unified spatiotemporal dependencies. FGN no longer treats input samples as a sequence of graphs; instead, it regards them as one complete graph. Therefore, Equation 2 can be rewritten as:

$$\hat{Y}_t^{(i)} := FGN_{\theta_g}(X_t^{(i)}, A_t^{(i)}), \tag{3}$$

where $X_t^{(i)} \in \mathbb{R}^{(T \times N) \times 1}$, $A_t^{(i)} \in \{1\}^{(T \times N) \times (T \times N)}$ is the adjacency matrix of a complete graph, and $\theta_g$ are the parameters of the FGN. FGN assumes complete connectivity between all sensor signals, with the adjacency matrix $A_t^{(i)}$ predetermined as a fully connected matrix (where every element is set to 1), thus avoiding the typical challenges associated with learning graph structures. In FGN, we initially project the node features into a higher dimensional space $d$ to obtain node embeddings $X_t^{(i)} \in \mathbb{R}^{(T \times N) \times d}$, and perform a Discrete Fourier Transform (DFT) to transform node embeddings into the frequency domain and get $\mathcal{F}(X_t^{(i)}) \in \mathbb{C}^{(\lfloor \frac{(T \times N)}{2} \rfloor + 1) \times d}$. Then we conduct recursive multiplications between $\mathcal{F}(X_t^{(i)})$ and Fourier Graph Operators (FGOs) in the Fourier space and make summations. Finally, we transform the node embeddings to the time domain using the Inverse Discrete Fourier Transform (IDFT), and utilize fully connected layers to map embeddings to labels, as illustrated in Figure 1. The detailed FGN procedure can be formulated as follows:

$$FGN_{\theta_g}(X_t^{(i)}, A_t^{(i)}) := \mathcal{F}^{-1}\left(\sum_{k=0}^{K} \sigma(\mathcal{F}(X_t^{(i)})S_{0:k} + b_k)\right), \quad S_{0:k} = \prod_{i=0}^{k} S_i, \tag{4}$$

where $\mathcal{F}(\cdot)$ and $\mathcal{F}^{-1}(\cdot)$ stand for DFT and IDFT, respectively. $S_k \in \mathbb{C}^{d \times d}$ is the FGO in the $k$-th layer. $\sigma$ is the activation function, and $b_k \in \mathbb{C}^d$ are the complex-valued bias parameters. By treating time series samples as complete graphs and performing transformations in the frequency domain, FGN effectively captures potential spatiotemporal inter-dependencies within sensor signal data, while eliminating the need for the graph structure learning phase typically required in conventional ST-GNNs.

## 3.3  Multi-term ensemble learning framework

### 3.3.1  Multi-term Training Process

When generating samples using the sliding time window method, one key parameter to set is the lookback window size, denoted as $T$. If only a single fixed window size is used, $T$ must not exceed the length of the shortest sequence in the dataset. Otherwise, some sequences would be excluded from the prediction and processing steps. This introduces the constraint $T \leq \min_{i \in \{1,2,...,M\}} L_i$, where $L_i$ represents the length of the $i$-th sequence. However, this constraint often forces the choice of a relatively small $T$, which may be insufficient to contain long-term dependencies in the data. To address this limitation, we propose a multi-term ensemble learning strategy. Instead of using a single lookback window, we employ multiple lookback windows to create a diverse set of training samples:

$$T = \{T_{\min}, T_{\min} + 1 \times D, \ldots, T_{\min} + (C-1) \times D\}, \tag{5}$$

where $C$ is the number of lookback windows, and $D$ is the increment in window size. The minimum lookback window $T_{\min}$ is set to be less than or equal to the shortest sequence length in the test dataset. This ensures

that predictions can be made for all test sequences. Next, we gradually enlarge the window size to gather extended long-term information. By utilizing various lookback windows of differing dimensions, we can generate multi-term samples, as depicted in Figure 1. This process results in $C$ groups of training samples, each corresponding to a different lookback window size, incorporating temporal dependencies at multiple scales. During the process of creating training samples, it is possible to encounter situations where the length of a training sequence is shorter than the designated lookback window size. In such cases, we discard the entire training sequence to ensure that all training samples maintain consistent input dimensions. Once the training samples are generated, we train $C$ individual FGN models, where each model is trained exclusively on the sample group generated by a specific lookback window size. This design allows each individual FGN to focus on capturing different inherent temporal patterns within the data, thereby enhancing the ensemble model's ability to learn from various time perspectives. By leveraging the complementary strengths of multiple FGNs, the overall performance of the ensemble model can be significantly improved. After training, we obtain a set of trained FGN models: $\{FGN_{trained}^{(1)}, FGN_{trained}^{(2)}, \ldots, FGN_{trained}^{(C)}\}$. Each of these models contributes uniquely to the final ensemble, making the model more robust and versatile in handling diverse time dependencies present in the data.

### 3.3.2 Adaptive Length-grouped Ensemble for Inference

Traditional methods commonly use data from the last time window of a test sequence as the test sample, which is then fed into a trained model to obtain a single prediction result. In contrast, our proposed framework leverages multiple lookback window sizes to generate multiple test samples from the end of each test sequence. These samples correspond to the same target, namely the RUL at the current point. Given that some shorter test sequences may not allow the generation of complete samples for larger time windows, our strategy is to group the test sequences according to their length and utilize applicable trained models for each group. Figure 1 shows the test set divided into two groups: long and short test sequences. For long test sequences, multiple samples can be generated using several lookback window sizes, and these samples are then input into the corresponding trained models, resulting in multiple prediction results. The extensive temporal information contained in long sequences allows them to leverage multiple models trained on diverse window sizes, facilitating the production of more robust and comprehensive predictions. Each prediction captures distinct temporal dependencies relevant to different time spans. In contrast, short sequences, which are more limited in the amount of historical data available, are processed using fewer models or, in certain cases, only a single model with the minimum window size. Let the set of test sequences be denoted by $\{X^{(1)}, X^{(2)}, \ldots, X^{(M)}\}$. For a specific test sequence $X^{(i)}$ with length $L_i$, the applicable set of lookback window sizes is determined as $T^{(i)} = \{T_k \in T \mid T_k \le L_i\}$. For each window size $T_j \in T^{(i)}$, a test sample is generated from the last $T_j$ time steps of the sequence $X^{(i)}$. This sample is subsequently fed into the corresponding pre-trained model, $FGN_{trained}^{(j)}$, to generate a prediction $\hat{y}_i^{(j)}$. This results in a collection of predictions given by: $\hat{\mathcal{Y}}_i = \{\hat{y}_i^{(1)}, \hat{y}_i^{(2)}, \ldots, \hat{y}_i^{(m)}\}$, where $m$ represents the number of valid window sizes for the sequence $X^{(i)}$. Finally, we employ a simple averaging to aggregate all the predictions, defined as: $\hat{y}_i = \frac{1}{m} \sum_{c=1}^{m} \hat{y}_i^{(l)}$, where $\hat{y}_i^{(l)}$ is the prediction from the $l$-th model.

This adaptive grouping strategy optimizes the utilization of available data for each test sequence while still adhering to the constraints imposed by the sequence length. It allows the ensemble of models to effectively capture both short- and long-term dependencies, improving the overall accuracy and stability of the predictions. By leveraging multiple trained models and dynamically selecting the applicable models for each test sequence, our framework adapts to the varying lengths of the test data, providing tailored predictions that are better suited to the inherent characteristics of each sequence.

### 3.4 Sequence decomposition under multiple operation conditions

Due to the equipment's varying operation conditions, the collected sensor signals often adhere to multiple distributions. A typical example of monitoring data with two operation conditions is illustrated in Figure 2. Sensor values frequently fluctuate between two distinct patterns, reflecting varying statistical properties and frequency components over time. Applying DFT directly to such mixed sequences may cause spectral aliasing, where the frequency components of each sequence interfere with one another. This results in overlapping

frequency information from multiple sequences, which diminishes the clarity and interpretability of the extracted frequency characteristics. Besides, abrupt transitions in the time series due to changing operation conditions may be mistakenly interpreted by the DFT as low-frequency components. These components are not truly indicative of equipment degradation but instead represent noise that should be eliminated. To mitigate these issues, we employ a simple sequence decomposition method to minimize interference from mixed signals and reduce noise. Our first step involves identifying operation conditions. If the condition monitoring data already includes records of operation conditions at each time point, we can analyze them directly. Alternatively, we extract information on operation conditions from sensor signals. We utilize the straightforward and efficient k-means (Ahmed et al., 2020) method for clustering analysis of the raw sensor signals to determine which operation condition a given sensor signal belongs to at timestep $t$. Then we segment the sensor signals based on the identified operation conditions. This involves dividing the data into subsets, each corresponding to a specific operation condition, as shown in the following formulation.

$$X^{(i)} \xrightarrow{\text{k-means}} \{X^{(i,1)}, X^{(i,2)}, \ldots, X^{(i,k)}\}, \tag{6}$$

where $X^{(i)}$ represents the $i$-th original time series, and $\{X^{(i,1)}, X^{(i,2)}, \ldots, X^{(i,k)}\}$ represent the time series resulting from the k-means clustering process, where $k$ is the number of clusters obtained by the k-means algorithm. The newly generated multiple time series data are complementary to each other, with many missing values that require interpolation. While numerous novel interpolation methods have been proposed (Oh et al., 2020), given the substantial number of data points requiring interpolation, we adopt a straightforward equal-value interpolation method, where missing values are filled using their nearby counterparts. The interpolation process can be formally expressed as:

$$\{X^{(i,1)}, X^{(i,2)}, \ldots, X^{(i,k)}\} \xrightarrow{\text{interpolation}} \{\hat{X}^{(i,1)}, \hat{X}^{(i,2)}, \ldots, \hat{X}^{(i,k)}\}. \tag{7}$$

Finally, we utilize a sliding time window approach to generate samples $\{\hat{X}_t^{(i,1)}, \hat{X}_t^{(i,2)}, \ldots, \hat{X}_t^{(i,k)}\}$ for each operation condition sequence, inputting them into FGNs. The predicted mean under different operation conditions is then the final output:

$$\hat{Y}_t^{(i)} := \frac{1}{k} \sum_{l=1}^{k} FGN_{\theta_g}^l(\hat{X}_t^{(l,k)}, \hat{A}_t^{(l,k)}). \tag{8}$$

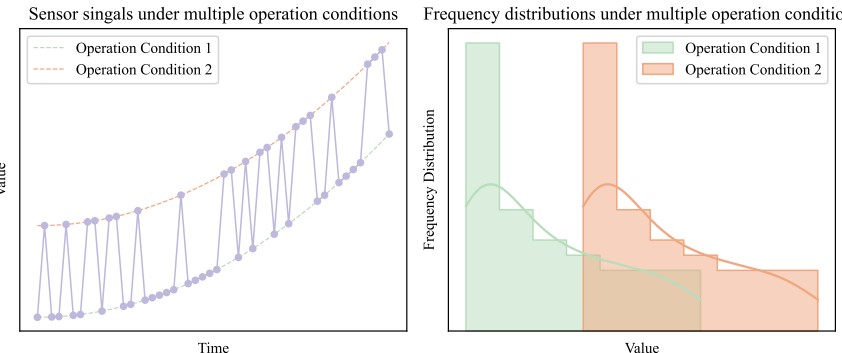

Figure 2: A typical example illustrates original sensor signals and their frequency distributions under multiple operation conditions.

## 4  Experiment and Results

In this section, we present a comprehensive evaluation of the proposed MT-FGNE framework on two widely used benchmark datasets for RUL prediction.

### 4.1 Datasets description and analysis

C-MAPSS: The Commercial Modular Aero-Propulsion System Simulation (C-MAPSS) dataset is a widely used public dataset in the field of RUL prediction (Xia et al., 2020). There are four subsets in this dataset. Each subset is divided into a training set and a test set. The training set contains multiple turbofan engine condition monitoring data from healthy operation to complete failure. The condition monitoring data in the test set ends before complete failure occurs. The goal is to forecast the engines' RUL in the test set. The characteristics of each subset are detailed in Table 1. Among the four data sets FD001-FD004, the engines in FD001 and FD003 operated under a single operation condition, while those in FD002 and FD004 operated under six different operation conditions, increasing the prediction complexity. Additionally, engines in FD001 and FD002 only have one fault mode, which is the High-Pressure Compressor (HPC) failure, while FD003 and FD004 hold two fault modes. Table 1 also displays the minimum and maximum sequence lengths of the dataset, indicating significant variations in sequence lengths across different engines. Since training data captures engine operation up to the point of failure, the signal record is usually relatively long. Also, variations in initial engine states and failure processes lead to different sequence lengths for each engine. Existing models (Kong et al., 2022; Wang et al., 2021; Chen & Zeng, 2023) commonly use a single fixed lookback window to generate samples. However, this lookback window size cannot exceed the minimum sequence length of the test engines; otherwise, the model cannot be applied to obtain predictions for all test engines. This relatively small lookback window size is inappropriate for test engines with extensive sensor data and may hinder the model's ability to learn long-term dependencies.

N-CMAPSS: the N-CMAPSS dataset is introduced by Arias Chao et al. (2021), it provides a more comprehensive and realistic simulation of aircraft engine operations. This dataset represents a significant advancement over C-MAPSS by incorporating complete flight phases (climb, cruise, and descent) and offering more complex, real-world operational scenarios, making it an excellent benchmark for assessing our model's robustness and generalization capabilities. The N-CMAPSS dataset contains eight sets of data from 128 units, and the DS02 subset is utilized for data-driven prognostics. The units in DS02 include two failure modes: one characterized by High-Pressure Turbine (HPT) efficiency degradation and the other involving combined Low-Pressure Turbine (LPT) efficiency and flow issues along with HPT degradation. Six units (2, 5, 10, 16, 18, and 20) serve as the training dataset, while units 11, 14, and 15 are allocated for testing purposes. As shown in Table 1, the dataset contains significantly long sequences due to high-frequency signal sampling within each flight cycle, providing sufficient data for flexible window size selection in sample generation.

Table 1: Description of C-MAPSS and N-CMAPSS turbofan engine dataset.

| Dataset | Subsets | Operation Condition | Fault Mode | Training units | Test units | Max length | Min length |
|---------|---------|---------------------|------------|----------------|------------|------------|------------|
| | FD001 | 1 | HPC | 100 | 100 | 362 | 31 |
| C-MAPSS | FD002 | 6 | HPC | 260 | 259 | 378 | 21 |
| | FD003 | 1 | HPC+Fan | 100 | 100 | 525 | 38 |
| | FD004 | 6 | HPC+Fan | 249 | 248 | 543 | 19 |
| N-CMAPSS | DS02 | - | HPT HPT+LPT | 6 | 3 | 1074k | 187k |

Table 2: Comparison of multiple time windows used in the proposed framework against existing single time windows.

| Dataset | Subsets | Min length | Single window size (Wang et al., 2021) | Window sizes in MT-FGNE |
|---------|---------|------------|----------------------------------------|-------------------------|
| | FD001 | 31 | 30 | 30/60/90/120/150/180 |
| C-MAPSS | FD002 | 21 | 20 | 20/40/60/80/100/120/140/160/180 |
| | FD003 | 38 | 30 | 30/60/90/120/150/180/210/240 |
| | FD004 | 19 | 15 | 18/40/62/84/106/128/150/172/194 |
| N-CMAPSS | DS02 | 1.87k | 50 | 100/200 |

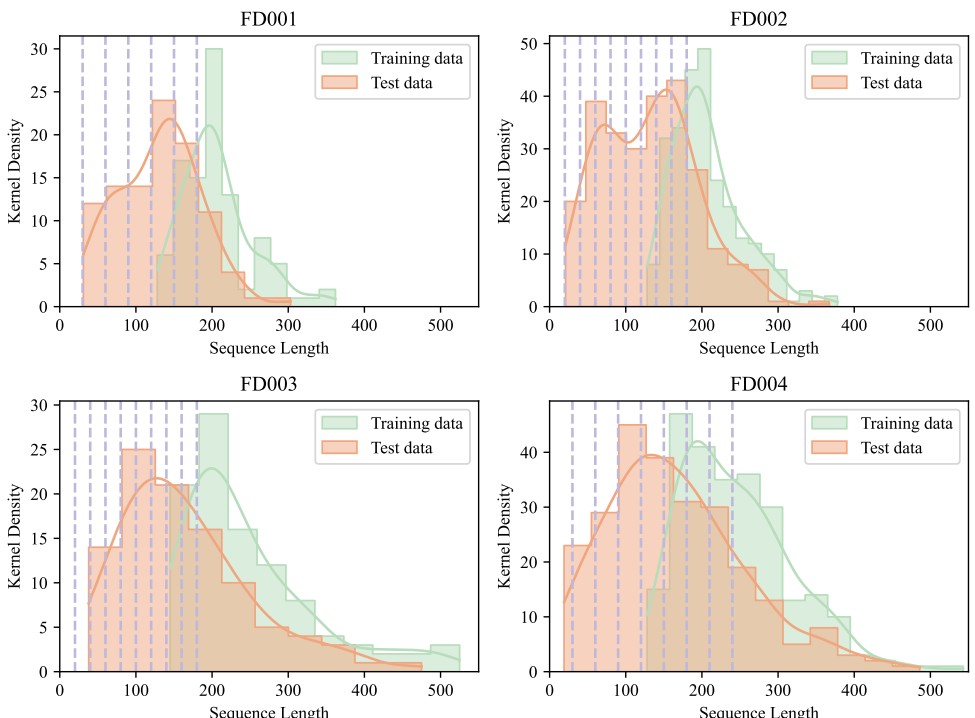

Figure 3: The training and test sequence length distribution of four subsets in the C-MAPSS dataset, multiple dashed lines parallel to the y-axis represent the various sizes of the lookback window we employed. These dashed lines partition the training and testing time series into multiple subgroups.

## 4.2 Implementation details

We maintain consistency in the data preprocessing settings as in (Kong et al., 2022; Wang et al., 2021) to ensure a fair comparison with existing models. For C-MAPSS, we first normalize the 14 effective features selected from the original 24 features. Then, we apply a piecewise function to rectify the training and test labels, ensuring that they do not exceed 125 to mitigate the possibility of overestimating RUL. Next, we apply the proposed multi-term learning approach by employing multiple lookback windows to generate various samples. For the four subsets, the lookback window sizes used are shown in Table 2. Unlike existing methods that use a single window size, which cannot exceed the minimum sequence length in the subset, resulting in short samples that fail to capture long-term dependencies in the time series, we adopt a set of lookback window sizes. We keep the first window size smaller than the minimum sequence length to ensure the applicability of the predictive model. Subsequently, we gradually increase the window size to generate longer samples, allowing the model to learn potential long-term dependencies. The employed lookback time windows and their comparison with the length of the time series are illustrated in Figure 3. We partition the training and testing sequences into subgroups using the multiple lookback windows we defined. We generate training and testing samples adaptively based on the length of sequences within each subgroup. Given the relatively longer sequences on the right side of each sub-figure, they could adopt all lookback windows on the left side of the sequences, while the reverse was not valid. For N-CMAPSS, the data is downsampled to a rate of 0.01 Hz, and a similar approach is employed to generate training and test samples. To guarantee a comprehensive and equitable comparison with the cutting-edge models cited in Wang et al. (2021), we evaluated our model on the complete degradation trajectories of individual test units as well as on the entire test dataset.

For C-MAPSS, the operation of FD002 and FD004 under six different conditions may result in a low signal-to-noise ratio when FGN is applied directly for frequency domain learning. To address this, we select six

features highly correlated with the labels $\{s7, s9, s11, s12, s13\}$ (Huang et al., 2023) and categorize the data into six distinct clusters. For N-CMAPSS, the raw signals are clustered into four groups based on four parameters of the operation condition: altitude, flight Mach number, Throttle Resolver Angle (TRA), and total temperature at the fan inlet (T2). After completing the clustering process, we apply a straightforward equal-value interpolation to address the missing values. Next, FGN is used to learn from each of the subsequences separately. We set the number of FGO layers to three, which is deep enough for the RUL prediction task. The proposed model's training parameters were optimized using the RMSprop optimizer, with the following hyperparameters: 100 epochs, a learning rate of 0.001, and a batch size of 256. Two metrics are employed in the evaluation, including the Root Mean Square Error (RMSE) and a Score function (Kong et al., 2022) as described in the following equation:

$$\text{Score}(v, \hat{v}_i) = \begin{cases} \sum_{i=1}^{M}(e^{-\frac{\hat{v}_i - v_i}{13}} - 1) & \text{if } \hat{v}_i < v_i; \\ \sum_{i=1}^{M}(e^{\frac{\hat{v}_i - v_i}{10}} - 1) & \text{if } \hat{v}_i \geq v_i, \end{cases} \tag{9}$$

where $v_i$ and $\hat{v}_i$ represent the true and predicted RUL values, respectively. The asymmetric Score function imposes a more significant penalty for overestimating RUL, as overestimating RUL leads to more severe consequences. Similarly to RMSE, a lower value of the Score function indicates a better prediction performance.

### 4.3 Comparisons with state-of-the-art

This section compares our method with the most advanced RUL prediction techniques available. We mainly compare our method with ST-GNNs because of their outstanding performance on this task. Table 3 and Table 4 present comparisons of various approaches on the C-MAPSS and N-CMAPSS datasets, respectively. The current advanced approaches can be broadly categorized into two groups. The first group consists of sequence models, with the most popular being Transformer-based models. The second group comprises ST-GNNs, which excel in predictive performance by capturing spatial information, generally outperforming sequence models. Our approach differs significantly from existing methods. First, FGN conducts learning in the frequency domain space instead of the time domain. Secondly, we transform samples into a graph rather than a sequence of graphs. Finally, we adopt a multi-term learning approach to enhance the model's learning of long-term dependencies within samples.

We compare our method with several state-of-the-art models on the C-MAPSS dataset. The values reported in Table 3 are taken directly from the papers as their implementations are not publicly available. One immediate observation on the performances of these baseline models is that there is no single best model that is capable of outperforming others on all four datasets, demonstrating the difficulty of the prediction task and the diversity shown in the datasets. In comparison to these baselines, our proposed MT-FGNE demonstrates superior performance on the first three datasets, FD001, FD002, and FD003. It shows a 13.6%, 2.3%, and 5.3% improvement compared to the second-best baseline on these three datasets, respectively. On FD004, our model is ranked second among eighteen models in terms of RMSE, with only small gaps to the best-performing models. The results indicate that even under various operation conditions and relatively low signal-to-noise ratios in original sensor signals, FGN can effectively learn dependencies after proper sequence decomposition and processing. Compared to traditional ST-GNNs, our method does not require learning graph structures or separately learning spatial and temporal information. Moreover, the Fourier Graph Operator is significantly less computationally expensive than the message-passing operators of GNNs. We follow STFA (Kong et al., 2022) and create an ST-GNN. The training time of ST-GNN and FGN on FD001 is 2518s and 729s, and the trainable parameters of ST-GNN and FGN are 183,905 and 74,064, respectively, showing that FGN has a lower computational burden. The total training time of MT-FGNE on FD001 is 5616s, which reflects the computational overhead of training multiple FGN models in the ensemble. Additionally, for FD002 and FD004, which require sequence decomposition, the processing time increases approximately six times due to the additional decomposition operations. Although this represents a notable increase in computational requirements compared to the base FGN, we believe this trade-off is justified by the significant improvements in prediction accuracy and data utilization rates demonstrated in our experimental results.

The experimental results shown in Table 4 demonstrate the superior performance of our proposed MT-FGNE compared to the state-of-the-art methods in different units of the N-CMAPSS dataset. Across different units,

our method consistently achieves lower Score metrics compared to baseline models, with significant reductions of up to 40% in Score values compared to transformer-based methods such as Transformer and Informer. Although LOGO (Wang et al., 2023b) achieves slightly better RMSE values in some cases, MT-FGNE demonstrates superior performance in terms of Score, which is a more comprehensive evaluation criterion for the N-CMAPSS dataset. Incorporating the N-CMAPSS dataset into our evaluation provides a thorough assessment of the model's generalization capabilities in realistic settings. Overall, our method consistently delivers outstanding performance across different scenarios, demonstrating its ability to provide significant contributions to RUL prediction.

Table 3: Compare the RMSE and Score values of MT-FGNE with other advanced sequence models and ST-GNN methods for the C-MAPSS dataset (bold: best; underline: runner-up).

| Models | FD001 | | FD002 | | FD003 | | FD004 | | Average | |
|---|---|---|---|---|---|---|---|---|---|---|
| | RMSE | Score | RMSE | Score | RMSE | Score | RMSE | Score | RMSE | Score |
| DA-Transformer (Liu et al., 2022) | 12.25 | 198 | 17.08 | 1575 | 13.39 | 290 | 19.86 | 1741 | 15.65 | 951.00 |
| BiGRU-TSAM (Zhang et al., 2022a) | 12.56 | 213 | 18.94 | 2264 | 12.45 | 233 | 20.47 | 3610 | 16.11 | 1580.00 |
| MSIDSN (Zhao et al., 2023) | 11.74 | 206 | 18.26 | 2047 | 12.04 | 196 | 22.48 | 2911 | 16.13 | 1340.00 |
| EAPN (Zhang et al., 2023) | 12.11 | 245 | 15.68 | 1127 | 12.52 | 267 | 18.12 | 2051 | 14.61 | 922.50 |
| Crossformer (Wang et al., 2023b) | 12.11 | 216 | 14.16 | 837 | 12.32 | 260 | 14.81 | 956 | 13.35 | 567.25 |
| HAGCN (Li et al., 2021) | 11.93 | 222 | 15.05 | 1144 | 11.53 | 240 | 15.74 | 1219 | 13.56 | 706.25 |
| STGCN (Wang et al., 2021) | 14.55 | 402 | 14.58 | 943 | 13.06 | 394 | 14.60 | 1065 | 14.20 | 701.00 |
| STFA (Kong et al., 2022) | 11.35 | 194 | 19.17 | 2493 | 11.64 | 225 | 21.41 | 2760 | 15.89 | 1418.00 |
| DAST (Zhang et al., 2022b) | 11.43 | 203 | 15.25 | 925 | 11.32 | **155** | 18.36 | 1491 | 14.09 | 693.50 |
| GGCN (Wang et al., 2022) | 11.82 | 187 | 17.24 | 1494 | 12.21 | 245 | 17.36 | 1372 | 14.66 | 824.50 |
| ConvGAT (Chen & Zeng, 2023) | 11.34 | 197 | 14.12 | 772 | 10.97 | 235 | 15.51 | 1231 | 12.99 | 608.75 |
| CDSG (Wang et al., 2023a) | 11.26 | 188 | 18.13 | 1740 | 12.03 | 218 | 19.73 | 2332 | 15.29 | 1119.50 |
| DCFA (Gao et al., 2023) | 11.74 | 190 | 16.81 | 1076 | 10.71 | 198 | 17.77 | 1571 | 14.26 | 758.75 |
| LOGO (Wang et al., 2023b) | 12.13 | 226 | 13.54 | 832 | 12.18 | 261 | **14.29** | **944** | 13.04 | 565.75 |
| NSD-TGTN (Gao et al., 2024) | 12.13 | 226 | 15.87 | 1477 | 12.01 | 220 | 16.64 | 1493 | 14.16 | 854.00 |
| DVGTformer (Wang et al., 2024) | 11.33 | 180 | 14.28 | 797 | 11.89 | 255 | 15.50 | 1108 | 13.25 | 585.00 |
| THGNN (Wen et al., 2024) | 13.15 | 285 | 13.84 | 806 | 12.61 | 255 | 14.65 | 1166 | 13.56 | 628.00 |
| MT-FGNE | **9.73** | **152** | **13.23** | **694** | **10.14** | 178 | 14.40 | 958 | **11.88** | **495.50** |

Table 4: Compare the RMSE and Score values of MT-FGNE with other advanced sequence models and ST-GNN methods for the N-CMAPSS dataset (bold: best; underline: runner-up).

| Models | Unit11 | | Unit14 | | Unit15 | | All | |
|---|---|---|---|---|---|---|---|---|
| | RMSE | Score | RMSE | Score | RMSE | Score | RMSE | Score |
| Transformer (Mo et al., 2021) | 5.86 | 7725 | 7.69 | 2397 | 5.34 | 3391 | 6.54 | 17075 |
| Informer (Zhou et al., 2021a) | 6.28 | 8019 | 7.67 | 2437 | 5.03 | 3195 | 6.24 | 16066 |
| Autoformer (Wu et al., 2021a) | 10.73 | 23987 | 13.72 | 9533 | 11.29 | 15383 | 11.48 | 50322 |
| Crossformer (Zhang & Yan, 2023) | 6.89 | 11816 | 8.58 | 2855 | 7.39 | 5733 | 6.87 | 16704 |
| DAG (Li et al., 2019) | 9.00 | 21282 | 9.37 | 4211 | 7.30 | 7970 | 8.82 | 36780 |
| STGCN (Wang et al., 2021) | 8.96 | 32080 | 11.96 | 17665 | 8.39 | 20169 | 9.31 | 37710 |
| HAGCN (Li et al., 2021) | 6.39 | 9956 | 7.03 | 2009 | 7.48 | 9376 | 6.67 | 17918 |
| MAGCN (Chen et al., 2023) | 8.33 | 13835 | 10.52 | 4566 | 5.11 | 3623 | 7.37 | 20821 |
| LOGO (Wang et al., 2023b) | **5.73** | 7509 | **6.72** | 1940 | **4.54** | 3017 | **6.07** | 15127 |
| MT-FGNE | 6.16 | **4711** | 8.12 | **1347** | 5.81 | **2388** | 6.32 | **8447** |

### 4.4 Ablation studies

To demonstrate the effectiveness of the proposed framework, we conducted an ablation study comparing MT-FGNE with its variants on the C-MAPSS dataset. We primarily evaluated two components of MT-FGNE: the multi-term ensemble learning (MTE) framework, and the sequence decomposition (SD) plugin.

#### 4.4.1 Multi-term ensemble ablations

The first variant is the single FGN model. Like existing methods, individual FGN adopts smaller, fixed time windows to generate samples at a certain scale, resulting in unsatisfactory predictive performance. As shown in Table 5, compared to individual FGN, our MT-FGNE reduced 18.0% in prediction error on the FD001 dataset. Similar effects are observed across all four datasets, demonstrating the high applicability of the proposed framework. As analyzed earlier, the multi-term ensemble learning framework primarily relies on generating longer samples to enhance the model's ability to learn long-term dependencies. The prerequisite for performance improvement is that the adopted model possesses strong sequence modeling capabilities. For instance, MT-FGNE w/o SD applied multi-term learning, yet its performance on FD002 did not show significant enhancement. It is attributed to the sensitivity of the adopted model to noise in sensor signals, leading to poor learning capabilities. Merely increasing the time window does not contribute to resolving the noise issue in this situation.

The proposed multi-term ensemble learning framework is not limited to a specific individual model for learning and theoretically can be applied to all existing models to further enhance performance. We employed a simple sequence modeling tool, CNN, as the individual model. Following the adoption of multi-term learning, its predictive performance saw significant improvement. Particularly noteworthy is its performance on the FD002 and FD004 datasets, where the prediction error surpassed that of MT-FGNE and even outperformed the current state-of-the-art model by 16.2%. This result suggests that employing smaller, fixed time window settings hinders the model's ability to capture long-range dependencies in the data, thereby limiting the performance of most existing models. Figure 4 illustrates the performance of individual FGN models and ensemble results on the last four test subsets of the FD001 dataset. In subset 3, the test sequence length is insufficient to apply all FGN models, while in subset 6, the sequence length exceeds the maximum time window, set at 180, thus allowing the application of all trained FGN models for prediction. FGN performs better in the majority of cases (24 out of 34, or 70%) when a larger time window is applied. A larger time window for generating samples enables the FGN to capture more extensive long-range dependencies. However, extending the time window also results in fewer generated samples, thereby impacting the model's training. Consequently, in some cases the prediction performance fails to improve or might even decline. In most cases, the ensemble predictions further enhance accuracy and significantly reduce the variance of prediction errors, indicating improved stability of the predictive models.

Table 5: Ablation study on the C-MAPSS dataset.

| Variants | FGN | MTE | SD | FD001 | | FD002 | | FD003 | | FD004 | |
|---|---|---|---|---|---|---|---|---|---|---|---|
| | | | | RMSE | Score | RMSE | Score | RMSE | Score | RMSE | Score |
| CNN | ✗ | ✗ | ✗ | 12.71 | 241 | 15.74 | 1504 | 11.47 | 185 | 17.81 | 1809 |
| MT-CNNE w/o SD | ✗ | ✓ | ✗ | - | - | **11.83** | **537** | - | - | **13.76** | **853** |
| MT-CNNE w/o MTE | ✗ | ✗ | ✓ | - | - | 17.21 | 1901 | - | - | 19.67 | 3124 |
| MT-CNNE | ✗ | ✓ | ✓ | 11.76 | 248 | 14.11 | 851 | 10.77 | 240 | 15.54 | 1147 |
| FGN | ✓ | ✗ | ✗ | 11.87 | 194 | 20.87 | 2221 | 12.64 | 208 | 27.22 | 4777 |
| MT-FGNE w/o SD | ✓ | ✓ | ✗ | - | - | 19.21 | 1563 | - | - | 23.38 | 2781 |
| MT-FGNE w/o MTE | ✓ | ✗ | ✓ | - | - | 16.39 | 1411 | - | - | 18.37 | 2168 |
| MT-FGNE | ✓ | ✓ | ✓ | **9.73** | **152** | 13.23 | 694 | **10.14** | **178** | 14.40 | 958 |

MTE: Multi-Term Ensemble, SD: Sequence Decomposition.

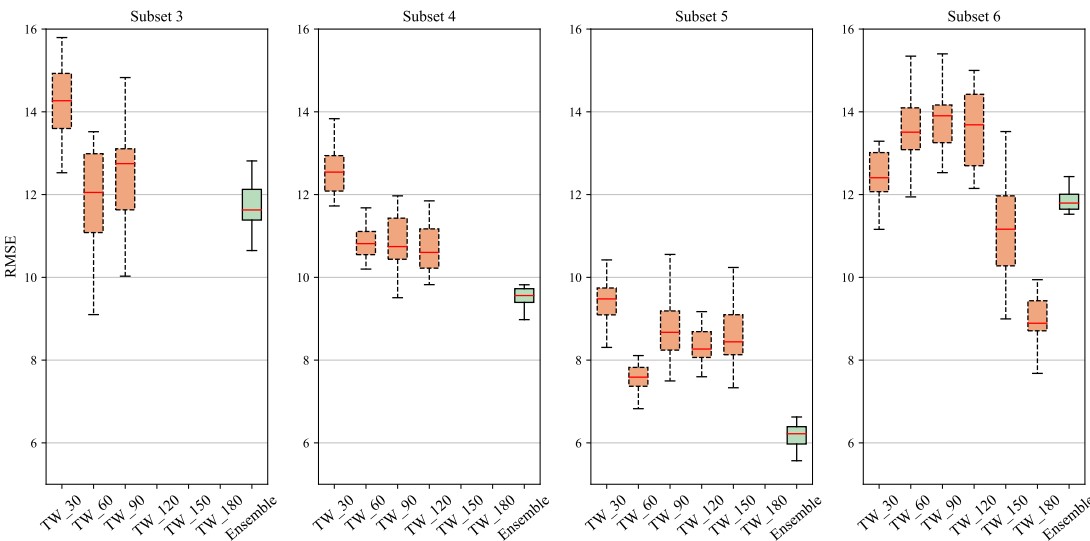

Figure 4: The individual FGN models' performance and ensemble results on the last four test subsets in FD001.

The proposed framework can also enhance the utilization of test data. We define the test data utilization rate as the ratio of the length of the time steps used (specifically, the final window for prediction) to the entire length of the test time series. The original FGN models exhibit relatively low utilization rates ranging from 9.03% to 22.91%, our MT-FGNE framework significantly increases these rates across all four benchmark datasets, achieving utilization rates between 80.65% and 88.32%. This marked improvement demonstrates our model's capability to effectively leverage a larger portion of the test sequences. A straightforward hypothesis suggests that similar results might be achieved by using a single model with a larger lookback time window. However, our empirical results point to a different outcome. We conducted comparative experiments on datasets FD001 and FD003 to empirically validate this hypothesis. As illustrated in Figure 5, simply increasing the time window size in the single FGN model leads to a consistent degradation in performance. This performance deterioration can be attributed to the necessary padding operations required for shorter sequences when using larger window sizes. The padding introduces artificial data points that do not represent actual sensor measurements or degradation patterns, thereby corrupting the model's ability to capture genuine temporal dependencies. In contrast, our proposed MT-FGNE effectively captures temporal dependencies without compromising prediction accuracy through artificial data padding.

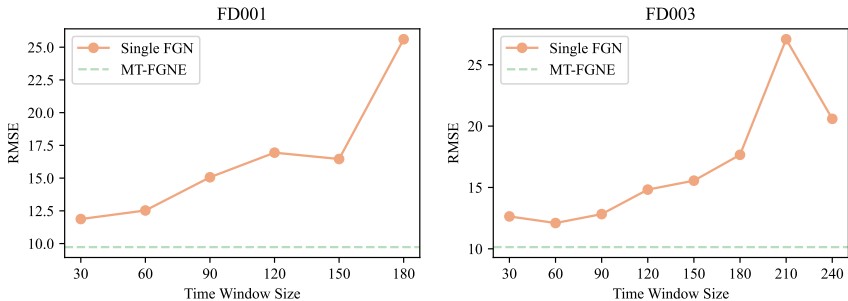

Figure 5: Performance comparison between Single FGN with larger time window sizes and MT-FGNE on FD001 and FD003 datasets.

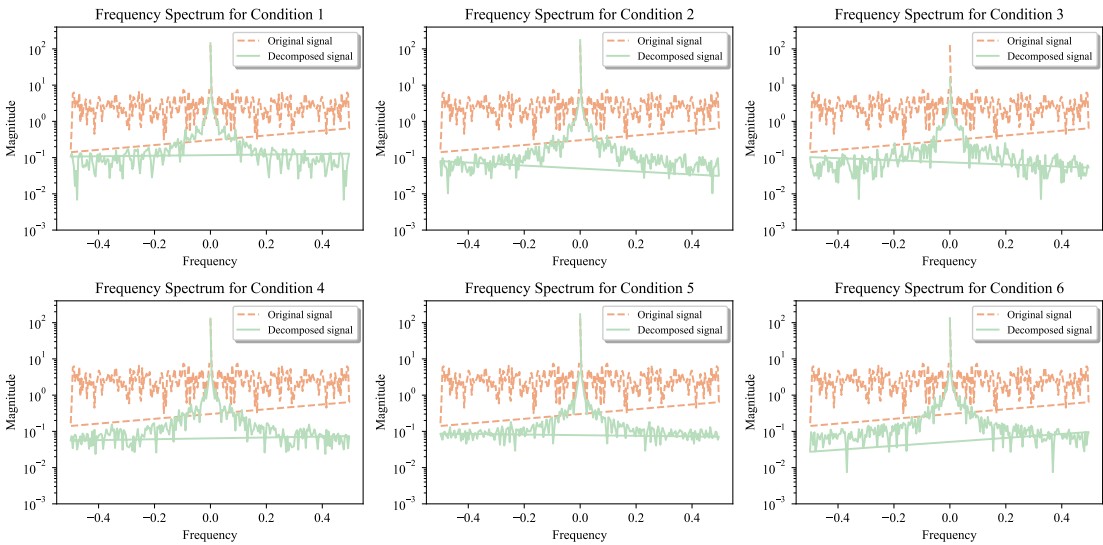

Figure 6: The frequency spectrum comparison of raw sensor signals and signals after decomposition in FD002.

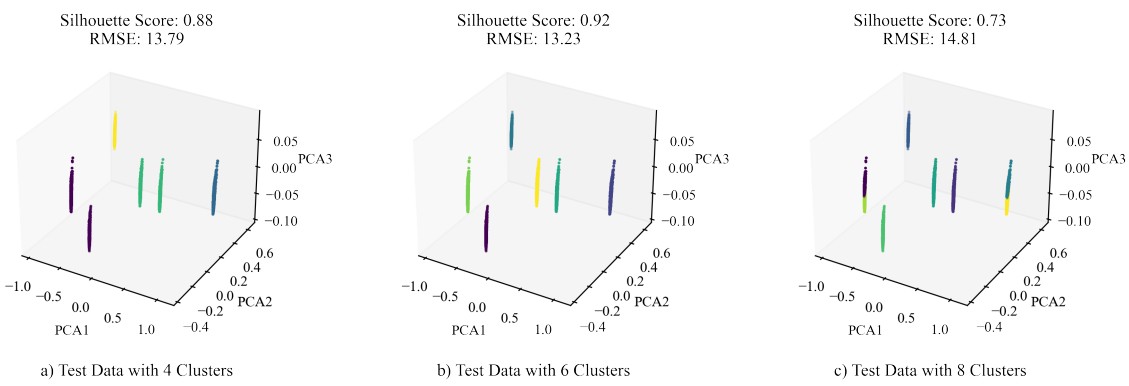

Figure 7: Comparison of clustering quality and prediction performance under different numbers of clusters (k=4, 6, 8) on the FD002 dataset.

### 4.4.2 Sequence decomposition ablations

The FD002 and FD004 datasets were generated under six different operation conditions. Our study primarily focuses on these two datasets to validate the effectiveness of sequence decomposition. As shown in Table 5, although the individual FGN w/o SD method exhibits decent predictive performance on FD001 and FD003, its predictive errors on FD002 and FD004 are quite significant. By comparing FGN with MT-FGNE w/o SD, we can find that the prediction performance of MT-FGNE w/o SD on FD002 is not significantly improved even with the MTE strategy. It indicates that even when the samples contain more information on long-term dependencies, the FGN model fails to learn from them effectively, diminishing the value of MTE. In contrast to FGN, the MT-FGNE w/o MTE model incorporates the SD plugin, resulting in a substantial improvement in predictive performance on FD002 and FD004. Figure 6 illustrates the frequency spectrum of an FD002 sample's original signal and the signal after denoising with SD. We observe a more uniform energy distribution in the original signal with frequent random fluctuations in frequency and no clear frequency range with high-amplitude signals. After denoising with SD, there is a significant increase in

amplitude in the low-frequency region of the frequency spectrum, indicating an enhancement in the signal-to-noise ratio. Finally, a significant enhancement in predictive performance is observed when combining SD for signal denoising, followed by the MTE module. This phenomenon underscores the necessity of applying SD for denoising signals, particularly for datasets with multiple operation conditions.

In real-world applications, the exact number of operation conditions is often unknown, posing a challenge for clustering-based SD methods. To address this, we utilized the Silhouette Score, a reputable clustering quality metric, to evaluate the SD quality and to detect potential unseen operation conditions. We conducted experiments on the FD002 dataset using different numbers of clusters, and examined Silhouette Score and the corresponding prediction performance. As shown in Figure 7, there are distinct and well-separated clusters for each condition, indicating that the k-means decomposition provides a clear segmentation of the operation conditions. Besides, we observed a strong correlation between clustering quality and model accuracy. Specifically, the configuration with six clusters achieved the highest clustering quality (Silhouette Score = 0.92) and the best prediction accuracy (RMSE = 13.23). Alternative configurations such as four clusters (Silhouette Score = 0.88, RMSE = 13.79) and eight clusters (Silhouette Score = 0.80, RMSE = 14.81) resulted in lower clustering quality and suboptimal predictive performance. These findings confirm that the SD approach can effectively capture underlying operation conditions, even when their exact number is unknown. Furthermore, by monitoring the Silhouette Score prior to training, we can detect potential unseen operational conditions through observed decreases in clustering quality, providing an early indication that the model configuration may require adjustment. This systematic approach offers a practical and robust solution for real-world scenarios, where operation conditions are often complex and unspecified.

## 5 Conclusion

This paper investigates the RUL prediction problem. Unlike existing deep learning models with fixed-size input samples, we consider the diversity in the time series lengths and propose a model capable of learning multi-term dependencies. We adaptively employ multiple lookback windows based on the time series length to generate multiple samples with the same label. Subsequently, these samples are transformed into complete graphs, and FGN models are employed to learn both spatial and temporal dependencies across multiple terms. The predictions from FGN models are then integrated to obtain the final prediction. This approach enhances the model's capability to learn long-range temporal dependencies in long sequences and further improves prediction performance by integrating predictions from different terms. For condition monitoring data generated under various operation conditions, we cluster the data based on operation conditions and then interpolate missing values, learn spectral features through FGN after DFT on data from different operation conditions, and finally average the prediction results. Extensive experimental results demonstrate that our proposed MT-FGNE model achieves state-of-the-art predictive performance on two turbofan engine datasets.

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
