# OpenReview forum: "Enhancing Remaining Useful Life Prediction with Ensemble Multi-Term Fourier Graph Neural Networks"
_TMLR — Accepted by TMLR_

### Review · Reviewer_ALmo · 2024-09-02

**Summary Of Contributions:**

This paper proposes a model name Ensemble Multi-Term Fourier Graph Neural Networks for Remaining Useful Life prediction. The proposed approach contains several major components: a Fourier Graph Neural Network to model the spatial-temporal dependencies within sensor data; a multi-term ensemble learning framework to learn both long-term and short-term temporal dependencies; a sequence decomposition method to tackle the varying operation conditions. The proposed method is evaluated on a widely-studied RUL prediction dataset and shows superior performance compared to several traditional and spatial-temporal GNN models. The motivations for each of these components are clear, and ablation studies are conducted to demonstrate the necessities of these components.

**Audience:**

Yes

**Broader Impact Concerns:**

This paper does not present significant ethical implications or potential negative societal impacts that makes a Broader Impact Statement necessary.

**Claims And Evidence:**

Yes

**Requested Changes:**

My main concerns are shown in the strengths and weaknesses section. Overall I think this is a paper of good quality. However, if the authors can provide more ablations on the choice of the network backbone, illustrate how different operation conditions influence the sensor signals and explain the results in $\underline{\text{Figure 4}}$ more clearly, I can be more confirmed towards acceptance of this paper.

**Strengths And Weaknesses:**

Strengths:

1. This paper is overall well written. The ideas are easy to follow.
2. The proposed multi-term ensemble learning framework is reasonable for capturing both long-term and short-term dependencies and boosting the model's performance.
3. The sequence decomposition method is a natural way to tackle the real-world scenarios where the sensor values switch between several distinct distributions. The data analysis in $\underline{\text{Section 3.4}}$ proves the existence of such problems.

Weaknesses:

1. In $\underline{\text{Section 4.4.1}}$, the results show a simple CNN model accompanied by multi-term learning and sequence decomposition perform better than the proposed method on dataset FD002 and FD004. While this shows that the multi-term framework and sequence decomposition approach is effective, the FGN model may be suboptimal for specific tasks. I suggest that ablations with regard to the network backbone should be conducted.
2. This paper utilizes k-means to determine which operation condition a signal belongs to. This may be somehow ad-hoc, as  k-means is effective to classify signals with different amplitudes into different clusters, as can be seen in $\underline{\text{Figure 2}}$. Is this always  the case? Maybe this paper can provide more analysis on how different operation conditions influence the final signals.
3. In $\underline{\text{Section 4.4.1}}$, the authors state that "FGN models trained with samples generated from larger time windows demonstrate superior predictive performance". However, if I’m not mistaken, the results in $\underline{\text{Figure 4}}$ seem to show that models trained with smaller windows can also show superior performance, for example, in Subset 3, "TW_60" shows better results than "TW_90", which is a little confusing for me.

---

> ### Author Response · Authors · 2024-09-24
> **Author Response (1/2)**
>
> Thank you for your constructive reviews! We appreciate your insight and respond below to your concerns:
>
> **Q1: Comparable performance by MT-CNNE.**
>
> **A1:** The ablation study reveals that the proposed Multi-Term Ensemble (MTE) framework can also work well with a CNN besides the Fourier Graph Neural Network (FGN). On datasets FD002 and FD004 with six distinct operational conditions, the CNN with MTE framework outperforms all current SOTA models, highlighting our MTE framework's general applicability. Meanwhile, we note that CNN performs less stable than FGN, with the largest Score on FD001 and FD003 (even larger than vanilla CNN), meaning severe overestimation of RUL. Therefore, the FGN performs generally better across diverse datasets.
>
> Following the reviewer's suggestion, we conducted additional ablation studies focusing on CNN, and the results are presented in the table below. Notably, MT-CNNE w/o SD performs better than MT-CNNE on FD002 and FD004. This demonstrates that the Sequence Decomposition (SD) component does not provide benefits for traditional models like CNN. The main reason might be that the signals exhibit greater stationarity after SD processing, aiding FGN's learning in the frequency domain. On the other hand, conventional temporal models like CNN concentrate on identifying local time patterns, making them proficient at learning features like signal amplitude variations.
>
> | Variants     	| FGN  | MTE  | SD   | RMSE (FD001) | Score (FD001) | RMSE (FD002) | Score (FD002) | RMSE (FD003) | Score (FD003) | RMSE (FD004) | Score (FD004) |
> |------------------|------|------|------|--------------|---------------|--------------|---------------|--------------|---------------|--------------|---------------|
> | CNN          	| ✗	| ✗	| ✗	| 12.71    	| 241       	| 15.74    	| 1504      	| 11.47    	| 185       	| 17.81    	| 1809      	|
> | MT-CNNE w/o SD   | ✗	| ✓	| ✗	| -        	| -         	| **11.83**	| **537**   	| -        	| -         	| **13.76**	| **853**   	|
> | MT-CNNE w/o MTE  | ✗	| ✗	| ✓	| -        	| -         	| 17.21    	| 1901      	| -        	| -         	| 19.67    	| 3124      	|
> | MT-CNNE      	| ✗	| ✓	| ✓	| 11.76    	| 248       	| 14.11    	| 851       	| 10.77    	| 240       	| 15.54    	| 1147      	|
> | FGN          	| ✓	| ✗	| ✗	| 11.87    	| 194       	| 20.87    	| 2221      	| 12.64    	| 208       	| 27.22    	| 4777      	|
> | MT-FGNE w/o SD   | ✓	| ✓	| ✗	| -        	| -         	| 19.21    	| 1563      	| -        	| -         	| 23.38    	| 2781      	|
> | MT-FGNE w/o MTE  | ✓	| ✗	| ✓	| -        	| -         	| 16.39    	| 1411      	| -        	| -         	| 18.37    	| 2168      	|
> | MT-FGNE      	| ✓	| ✓	| ✓	| **9.73** 	| **152**   	| 13.23    	| 694       	| **10.14**	| **178**   	| 14.40    	| 958       	|
>
> *Note: MTE: Multi-Term Ensemble, SD: Sequence Decomposition.*
>
> The sequence decomposition is still meaningful in real applications. Instead of training a complex model to handle all scenarios, relatively simple models can be developed for each specific operation condition. This approach helps reduce the risk of overfitting. In addition, by segmenting the data, it becomes easier to understand how different operation conditions influence the outcomes. This clarity can help make better-informed business decisions. Finally, in practical applications, the operation conditions can vary. Employing separate models for each condition allows for more flexibility in adjusting and optimizing models to adapt to new situations. In summary, we are convinced that FGN, which aligns effectively with the proposed MTE and SD frameworks, is the superior option.

---

> > ### Author Response · Authors · 2024-09-24
> > **Author Response (2/2)**
> >
> > **Q2: How different operation conditions influence the final signals.**
> >
> > **A2:** When the equipment functions under fluctuating conditions, the amplitudes of its sensor signals frequently show notable variations. For instance, in a turbofan engine operating at different altitudes, Mach numbers, and ambient temperatures, the amplitudes of temperature and pressure sensor signals can differ greatly, producing a signal pattern akin to that illustrated in `Figure 2`. If the signal amplitudes remain largely unchanged across different operation conditions, the signal can be viewed as stationary. In such cases, we can skip SD and move straight to Discrete Fourier Transform (DFT).
> >
> > To evaluate the effectiveness of the k-means method used for sequence decomposition in the proposed framework, we conducted parameter sensitivity analysis experiments on FD002.
> >
> > First, we tested the predictive performance under different numbers of clusters (*n_cluster*). The results showed that even with `n_cluster = 4` (RMSE = 13.79) and `n_cluster = 8` (RMSE = 14.81), the prediction error of MT-FGNE remained significantly lower than that of MT-FGNE w/o SD (RMSE = 19.21). This demonstrates that SD utilizing k-means remains reliable and efficient, even when the number of operation conditions is not predetermined. We further evaluated the clustering quality of the k-means method under different *n_cluster* values by calculating the Silhouette Coefficient (the Silhouette Coefficient ranges from -1 to 1, with values closer to 1 indicating better clustering performance). The results indicated that compared to `n_cluster = 4` (silhouette_score = 0.88, RMSE = 13.79) and `n_cluster = 8` (silhouette_score = 0.80, RMSE = 14.81), the default setting `n_cluster = 6` (silhouette_score = 0.92, RMSE = 13.23) yielded the highest clustering quality and the lowest prediction error. This suggests that the improvement in predictive performance brought by SD is positively correlated with clustering quality.
> >
> > Subsequently, we evaluated the predictive performance of all models on the sub-sequences corresponding to various operation conditions after decomposition. Overall, the predictive precision across the sub-sequences was consistent (RMSE = 15.59 ± 1.56), indicating that no particular operation condition is more essential than the others. Remarkably, the prediction using sub-sequences (RMSE = 15.59) substantially exceeded that based on the original data (RMSE = 21.01). Moreover, simple averaging further reduced the prediction error significantly (RMSE = 10.84), showcasing the SD method's efficiency.
> >
> > **Q3: FGNs perform better with larger time windows.**
> >
> > **A3:** This experimental conclusion is indeed imprecise. As shown in `Figure 4`, FGN performs better in the majority of cases (24 out of 34, or 70%) on the last four test subsets of FD001 when a larger time window is applied. A larger time window for generating samples enables the FGN to capture more extensive long-range dependencies. However, extending the time window also results in fewer generated samples, thereby impacting the model's training. Consequently, it is reasonable that in some cases the prediction performance fails to improve or might even decline.
> >
> > `Figure 4` also shows that the configuration of the time window plays a crucial role in influencing forecasting accuracy. Unlike the conventional approach of manually setting a single lookback window, our suggested multi-term sample generation approach notably enhances the model's performance in time series predictions.
> >
> > We have updated the manuscript according to the reviewers' comments, with the changes highlighted using a yellow background.

---

### Review · Reviewer_iwPa · 2024-10-09

**Summary Of Contributions:**

This paper proposes Ensemble Multi-Term Fourier Graph Neural Networks (MT-FGNE) for remaining useful life (RUL) prediction. MT-FGNE treats data as complete graphs and employs a Fourier Graph Neural Network (FGN) to model the graph in the frequency space. It also creates training and test graphs with varying sizes of lookback windows and uses multiple predictions for the ensemble. Additionally, MT-FGNE designs sequence decomposition for multiple operation conditions. The proposed method is evaluated on the C-MAPSS benchmark and outperforms some state-of-the-art approaches.

**Audience:**

Yes

**Broader Impact Concerns:**

No ethical concerns.

**Claims And Evidence:**

No

**Requested Changes:**

Please refer to the Strengths And Weaknesses part. This submission needs improvements by adding more significant technical contributions, more clarifications on some statements, and experiments on more large-scale datasets.

**Strengths And Weaknesses:**

Strengths:

* The writing of this paper is clear and the design of the method is easy to follow. The experiments also show some good performance compared with some SOTA approaches.

Weaknesses:

* The overall technical depth of the proposed method is limited. (a) Since the Fourier Graph Neural Network has already been proposed and this paper directly uses it to model the samples, the contribution in this paper is unclear. (b) The multi-term ensemble learning framework simply uses predictions from multiple lookback windows for the ensemble. This is more like a trick of data pre-processing, rather than a technical contribution in machine learning. It also makes the comparison between MT-FGNE and other baseline methods unfair, as these baselines receive less input information. (c) The motivation for decomposing operation conditions is not clear enough. Is it because that different conditions have different amplitudes? Can this be solved by data normalization? The designed method is a simple utilization of K-means. Why can we use K-means to separate these conditions?

* Some other statements and claims are not fully supported. (a) It is unknown why the proposed method can solve the issue of ‘need to learn the graph structure’. (b) It does not make sense to claim that existing methods ‘struggle to capture long-term dependencies’ because ‘window sizes are too small’. Can they simply use a larger window size to solve the problem?

* Some designs in the method, including treating data as a complete graph, using multi-term ensemble, and performing K-means to decompose different operation conditions may all introduce additional computational costs, which are not fully discussed in this paper. Besides, does this method need multiple terms of data in inference? Will this limit its practical application?

* The experimental evaluation is also a bit limited. MT-FGNE is only evaluated on one benchmark, and its size is small as shown in Table 1.

---

> ### Author Response · Authors · 2024-11-08
> **Author Response (1/5)**
>
> Thank you for your critical reviews! We appreciate your insight and respond below to your concerns:
>
> **Q1.a: FGN has already been proposed and the contribution is unclear.**
>
> **A1.a:** In the proposed MT-FGNE framework, we employ FGN as the base model. To the best of our knowledge, this is the first time that models like FGN, which represents the time sequence in the frequency domain, have been applied to the field of RUL prediction. As demonstrated in `Table 4`, directly applying FGN does not outperform current state-of-the-art models and performs particularly poorly when the time series includes multiple operation conditions. However, by employing the proposed MTE framework, we significantly improved FGN's prediction accuracy across four datasets, reducing the prediction error by 18.03%, 36.60%, 19.78%, and 47.10%, respectively. The performance improvement is substantial, with our model outperforming the current best models on three datasets while achieving second-best performance on the remaining one.
> Furthermore, our MTE framework is versatile and can incorporate any learning model as the base learner. Even when a simple CNN is used as the base model, the MTE framework still achieves remarkable performance. In particular, when the time series includes multiple operation conditions, the CNN with the MTE framework (MT-CNNE) outperforms the current best model by reducing prediction errors by 12.63% and 3.71%, respectively. This demonstrates the generalization capability of the proposed framework.
> We hope this explanation clarifies the novelty and effectiveness of our approach.
>
> **Q1.b: MTE is data pre-processing and the comparison with baseline methods is unfair.**
>
> **A1.b:** The proposed MTE framework represents a specialized ensemble learning strategy. During the training phase, various base models are trained with samples generated under varying lookback windows, allowing them to capture temporal dependencies at multiple scales. This is not just a data preprocessing technique but a time series modeling strategy. By training across multiple time windows, the MTE framework is able to model different temporal patterns more effectively. Furthermore, in the inference phase, we designed an Adaptive Length-grouped Ensemble strategy. This approach groups test sequences based on their length, allowing each sequence to generate a sufficient number of test samples, which are then input into the applicable trained models, generating multiple predictions that are then aggregated to form the final output.
> One of the key objectives of the MT-FGNE framework is to make more efficient use of available test data. Traditional models with a single fixed lookback window often have to use a small window size to ensure compatibility with short test sequences, which leads to underutilization of the available data. In contrast, our proposed framework significantly improves the utilization of test data. To illustrate this, we computed the test data utilization rate, defined as the average proportion of time points used for predictions relative to the test sequence length. As shown in the following table, our model achieves a significant improvement in this metric.
>
> | Model       | FD001  | FD002  | FD003  | FD004  |
> |-------------|--------|--------|--------|--------|
> | **ST-GNN**     | 22.91% | 15.24% | 18.08% | 9.03%  |
> | **MT-FGNE** | 86.82% | 88.32% | 84.42% | 80.65% |
>
> It is important to emphasize that our framework does not introduce any additional data. Instead, it optimizes the utilization of the available test data through an efficient modeling strategy. Thus, the total amount of training data and test data used in our approach is fundamentally the same as that used by the baseline models. Consequently, this optimization is part of the model design and does not violate any fairness principles.
>
> To address the reviewer’s concerns, we have revised the description of the MTE framework in the manuscript to clarify these points. Additionally, we include an analysis of test data utilization rates in the manuscript to highlight the significant difference in data usage between our framework and the baseline models.

---

> > ### Author Response · Authors · 2024-11-08
> > **Author Response (2/5)**
> >
> > **Q1.c: The motivation for decomposing operation conditions is unclear and might be solvable with data normalization.**
> >
> > **A1.c:** The motivation for decomposing operation conditions stems from the fact that signals may exhibit distinct patterns or features (such as frequency, amplitude, etc.) under different operation conditions. When applying Discrete Fourier Transform (DFT) directly to the original mixed sequence, the frequency components of each sequence can interfere with one another, resulting in an overlay of frequency information from multiple sequences, thereby reducing the clarity and interpretability of the frequency characteristics. Besides, abrupt transitions in the time series due to changing operation conditions may be mistakenly interpreted by the DFT as low-frequency components. These components are not truly indicative of equipment degradation but instead represent noise that should be eliminated. To mitigate these issues, we employ a simple Sequence Decomposition (SD) method to minimize interference from mixed signals and reduce noise.
> >
> > `Figure 5` provides an illustrative example, showing the significant difference between the spectrum before and after decomposition. In the ablation study, the performance difference between the FGN model with and without SD was substantial. This highlights the importance of this step in improving model performance. Moreover, this decomposition allows for a more granular understanding of the equipment's behavior under varying states, facilitating a clearer interpretation of the device's characteristics across different operation conditions.
> >
> > Simple data normalization is insufficient to address this issue. While normalization adjusts the amplitude of the data, it does not alter the underlying distribution or clustering patterns associated with different operation conditions. As a result, the frequency components of data collected under various conditions remains overlaid, which hinders the FGN model’s ability to learn effectively. As shown in `Table 4`, despite we applied data normalization to the input in the ablation study for the MT-FGNE w/o SD variant, its predictive performance was still significantly lower than that of the full MT-FGNE model.
> >
> > To address the reviewer’s concerns, we have revised the description of the SD method in the manuscript to provide a more detailed explanation.
> >
> > **Q1.d: The use of k-means for separating conditions is not fully justified.**
> >
> > **A1.d:** Different operation conditions typically result in varying feature distributions. The k-means algorithm is well-suited for capturing these variations in data arising from different operation conditions. The underlying assumption of k-means is that data points are distributed around certain centroids. Since the distribution of data centers can shift under different operation conditions, k-means helps to distinguish between these conditions by clustering the data according to these varying centroids. Additionally, we explored alternative clustering methods, such as Gaussian Mixture Models and DBSCAN. However, our experimental results indicate that while these methods offer comparable performance, k-means achieves similar results and also has the advantage of simplicity and computational efficiency.
> >
> > To address the reviewer's concerns regarding the use of k-means for SD, we conducted a detailed quality analysis of the approach. The results demonstrate that SD using k-means remains both reliable and efficient, even in scenarios where the number of operation conditions is not predefined. Moreover, we visualized the clustering results obtained by k-means, which clearly show that this straightforward method is sufficient to effectively separate the data generated under different operation conditions.
> >
> > **Q2.a: The method's solution to the graph structure learning issue is unclear.**
> >
> > **A2.a:** In traditional Spatial-Temporal Graph Neural Networks (ST-GNNs), the relationships between sensor signals within a single time slice are typically unknown. As a result, it is common to use graph structure learning techniques to search for an optimal adjacency matrix that can be fed into the GNN for message passing. However, in the FGN model we employ, we treat the entire sample as a complete graph, as described in Section 3.2. This means that we assume full connectivity among all sensor signals, where the adjacency matrix is predefined as a matrix consisting entirely of ones. Consequently, we bypass the need to search for or optimize an adjacency matrix. This not only simplifies the model by eliminating the graph structure learning step, but it also ensures that potential spatiotemporal interdependencies within the sensor signal data are fully captured.
> >
> > To address the reviewer’s concerns, we have updated Section 3.2 to provide a clearer explanation of how the FGN model handles the input samples to skip graph structure learning.

---

> > > ### Author Response · Authors · 2024-11-08
> > > **Author Response (3/5)**
> > >
> > > **Q2.b: Claiming that existing methods "struggle to capture long-term dependencies" due to "small window sizes" is flawed, as increasing the window size could easily address this issue.**
> > >
> > > **A2.b:** While it might seem intuitive to increase the window size directly to better capture long-term dependencies, this approach is often not feasible in practice due to the variability in sequence lengths across datasets. In existing methods, the window size is often constrained by the length of the shortest sequence. If we increase the window size beyond this limit, many sequences would not have enough data to fill the window and generate valid samples for training or inference. A possible solution is padding the shorter sequences to match the longer window size. However, padding adds artificial data to the sequences, which do not reflect the actual sensor readings and degradation patterns, leading to reduced predictive performance.
> > >
> > > Our approach avoids the complications of padding by using a multi-term strategy, which generates samples with varying window sizes that correspond to the available sequence lengths. This ensures that both short-term and long-term dependencies can be captured without the need to artificially extend shorter sequences. Our method is therefore better suited to handling the inherent variability in time series lengths, preserving the integrity of the data while still modeling long-term dependencies effectively.
> > >
> > > To address the reviewer's concern, we have added an ablation study to empirically evaluate the effect of simply increasing the window size and padding the short test sequences. The results clearly demonstrate that larger window sizes lead to higher prediction errors. The increase in error is directly related to the amount of padding required as the window size grows. Padding dilutes the meaningful temporal information and distorts the actual degradation patterns in the time series. In fact, the more padding is applied, the greater the prediction error becomes, as the model is forced to process more artificial data points that do not reflect the real dynamics of the system.
> > >
> > > Thus, simply increasing the window size does not effectively solve the problem of capturing long-term dependencies. Instead, it introduces additional challenges, such as increased artificial data, which compromises the integrity of the model’s predictions. Our approach, which uses multiple window sizes without padding, is specifically designed to address these issues by balancing the need for long-term dependency capture while maintaining accuracy.
> > >
> > >
> > > **Q3.a: Additional computational costs from designs are not fully discussed.**
> > >
> > > **A3.a:** We acknowledge the importance of discussing computational costs comprehensively. In our current manuscript, we initially compared the computational efficiency between the base FGN and ST-GNN models. Specifically, on the FD001 dataset, FGN demonstrates superior efficiency with a training time of 729s and 74,064 trainable parameters, compared to ST-GNN's 2518s and 183,905 parameters. This comparison shows that the base FGN architecture has a significantly lower computational burden.
> > >
> > > To address the reviewer's concern, we have expanded our discussion of computational costs to include the full MT-FGNE ensemble framework. The total training time for MT-FGNE on FD001 is 5616s, which reflects the computational overhead of training multiple FGN models in the ensemble. Additionally, for datasets FD002 and FD004, which require sequence decomposition, the processing time increases by approximately six-fold due to the additional decomposition operations. While this represents a notable increase in computational requirements compared to the base FGN, we believe this trade-off is justified by the significant improvements in prediction accuracy and data utilization rates demonstrated in our experimental results.

---

> > > > ### Author Response · Authors · 2024-11-08
> > > > **Author Response (4/5)**
> > > >
> > > > **Q3.b: The method may require multiple terms of data during inference, potentially limiting its practical application.**
> > > >
> > > > **A3.b:** We acknowledge that our proposed framework indeed requires multiple terms of data during inference. To ensure the model’s practical usability, we have implemented an Adaptive Length-grouped Ensemble for Inference strategy. Given that some short test sequences may not allow the generation of complete samples for large time windows, our strategy is to group the test sequences according to their length and utilize applicable, trained models for each group. For long test sequences, multiple samples can be generated using several lookback window sizes, and these samples are then input into the corresponding trained models, resulting in multiple prediction results. In contrast, short sequences, which are more limited in the amount of historical data available, are processed using fewer models or, in certain cases, only a single model with the minimum window size.
> > > >
> > > > This adaptive strategy ensures that our method does not require additional test data during inference but rather maximizes the utility of available historical information. By accommodating varying sequence lengths, our framework offers greater flexibility compared to traditional models with fixed-size inputs. This makes our approach particularly suitable for real-world applications where data availability may vary significantly.
> > > >
> > > > We have updated the manuscript with a detailed description of this inference strategy to illustrate its practical implementation and benefits better.
> > > >
> > > > **Q4: The experimental evaluation is limited, as only one small benchmark is used.**
> > > >
> > > > **A4:** We acknowledge the reviewer's valid concern about the breadth of experimental validation. To address this limitation, we have extended our experiments to include the N-CMAPSS dataset introduced by Chao et al. [1], which provides a more comprehensive and realistic simulation of aircraft engine operations. This dataset represents a significant advancement over C-MAPSS by incorporating complete flight phases (climb, cruise, and descent) and offering more complex, real-world operational scenarios, making it an excellent benchmark for assessing our model's robustness and generalization capabilities.
> > > >
> > > > We followed the experimental setup outlined in the literature, adopting the data split methodology from Chao et al. [1] and the detailed settings from Wang et al. [2]. Specifically, we utilized six units (2, 5, 10, 16, 18, and 20) for training, while units 11, 14, and 15 served as the test set. To ensure a thorough and fair comparison with the state-of-the-art models presented in Wang et al. [2], we evaluated our model on the complete degradation trajectories of individual test units as well as on the entire test dataset. The experimental results shown in the table below demonstrate the superior performance of our proposed MT-FGNE compared to state-of-the-art methods across different units on the N-CMAPSS dataset. Across different units, our method consistently achieves lower score metrics compared to baseline models, with significant reductions of up to 40% in score values compared to transformer-based methods like Transformer and Informer. Although LOGO achieves slightly better RMSE values in some cases, MT-FGNE demonstrates superior performance in terms of the score metric, which is a more comprehensive evaluation criterion for the N-CMAPSS dataset. In summary, incorporating the N-CMAPSS dataset into our evaluation provides a thorough assessment of the model's generalization capabilities in realistic settings, demonstrating its applicability to complex, real-world degradation modeling tasks.
> > > >
> > > > | Models | Unit11 | | Unit14 | | Unit15 | | All | |
> > > > |--------|---------|---------|---------|---------|---------|---------|---------|---------|
> > > > | | RMSE | Score | RMSE | Score | RMSE | Score | RMSE | Score |
> > > > | Transformer [3] | 5.86 | 7725 | 7.69 | 2397 | 5.34 | 3391 | 6.54 | 17075 |
> > > > | Informer [4] | 6.28 | 8019 | 7.67 | 2437 | _5.03_ | 3195 | _6.24_ | 16066 |
> > > > | Autoformer [5] | 10.73 | 23987 | 13.72 | 9533 | 11.29 | 15383 | 11.48 | 50322 |
> > > > | Crossformer [6] | 6.89 | 11816 | 8.58 | 2855 | 7.39 | 5733 | 6.87 | 16704 |
> > > > | DAG [7] | 9.00 | 21282 | 9.37 | 4211 | 7.30 | 7970 | 8.82 | 36780 |
> > > > | STGCN [8] | 8.96 | 32080 | 11.96 | 17665 | 8.39 | 20169 | 9.31 | 37710 |
> > > > | HAGCN [9] | 6.39 | 9956 | _7.03_ | 2009 | 7.48 | 9376 | 6.67 | 17918 |
> > > > | MAGCN [10] | 8.33 | 13835 | 10.52 | 4566 | 5.11 | 3623 | 7.37 | 20821 |
> > > > | LOGO [2] | **5.73** | _7509_ | **6.72** | _1940_ | **4.54** | 3017 | **6.07** | _15127_ |
> > > > | MT-FGNE | _6.16_ | **4711** | 8.12 | **1347** | 5.81 | **2388** | 6.32 | **8447** |

---

> > > > > ### Author Response · Authors · 2024-11-08
> > > > > **Author Response (5/5)**
> > > > >
> > > > > Reference:
> > > > >
> > > > > [1] Arias Chao, M., Kulkarni, C., Goebel, K., & Fink, O. (2021). Aircraft engine run-to-failure dataset under real flight conditions for prognostics and diagnostics. Data, 6(1), 5.
> > > > >
> > > > > [2] Wang, Y., Wu, M., Jin, R., Li, X., Xie, L., & Chen, Z. (2023). Local–global correlation fusion-based graph neural network for remaining useful life prediction. IEEE Transactions on Neural Networks and Learning Systems.
> > > > >
> > > > > [3] Mo, Y., Wu, Q., Li, X., & Huang, B. (2021). Remaining useful life estimation via transformer encoder enhanced by a gated convolutional unit. Journal of Intelligent Manufacturing, 32(7), 1997-2006.
> > > > >
> > > > > [4] Zhou, H., Zhang, S., Peng, J., Zhang, S., Li, J., Xiong, H., & Zhang, W. (2021, May). Informer: Beyond efficient transformer for long sequence time-series forecasting. In Proceedings of the AAAI conference on artificial intelligence (Vol. 35, No. 12, pp. 11106-11115).
> > > > >
> > > > > [5] Wu, H., Xu, J., Wang, J., & Long, M. (2021). Autoformer: Decomposition transformers with auto-correlation for long-term series forecasting. Advances in neural information processing systems, 34, 22419-22430.
> > > > >
> > > > > [6] Zhang, Y., & Yan, J. (2023, May). Crossformer: Transformer utilizing cross-dimension dependency for multivariate time series forecasting. In The eleventh international conference on learning representations.
> > > > >
> > > > > [7] Li, J., Li, X., & He, D. (2019). A directed acyclic graph network combined with CNN and LSTM for remaining useful life prediction. IEEE Access, 7, 75464-75475.
> > > > >
> > > > > [8] Wang, M., Li, Y., Zhang, Y., & Jia, L. (2021). Spatio-temporal graph convolutional neural network for remaining useful life estimation of aircraft engines. Aerospace Systems, 4(1), 29-36.
> > > > >
> > > > > [9] Li, T., Zhao, Z., Sun, C., Yan, R., & Chen, X. (2021). Hierarchical attention graph convolutional network to fuse multi-sensor signals for remaining useful life prediction. Reliability Engineering & System Safety, 215, 107878.
> > > > >
> > > > > [10] Chen, L., Chen, D., Shang, Z., Wu, B., Zheng, C., Wen, B., & Zhang, W. (2023). Multi-scale adaptive graph neural network for multivariate time series forecasting. IEEE Transactions on Knowledge and Data Engineering, 35(10), 10748-10761.
> > > > >
> > > > > We hope that our responses have solved your problems. Thank you for your thoughtful comments again.

---

### Review · Reviewer_9m7r · 2024-10-23

**Summary Of Contributions:**

The paper introduces an Ensemble Multi-Term Fourier Graph Neural Network (MT-FGNE) model aimed at improving Remaining Useful Life (RUL) prediction by leveraging Fourier transforms to capture spatiotemporal dependencies. The proposed approach offers a novel solution to model complex interactions between spatial and temporal data while adapting to varying operational conditions. The authors claim that their method addresses key shortcomings in existing models and demonstrates superior performance on standard RUL prediction datasets.

**Audience:**

Yes

**Broader Impact Concerns:**

No significant ethical concerns are apparent in the paper’s current scope. However, it would be beneficial to add a broader impact section discussing the potential real-world implications of improved RUL prediction models in industries where predictive maintenance is critical, such as aviation and manufacturing. Specifically, attention should be given to ensuring that such models do not inadvertently lead to over-reliance on predictive systems, which could introduce new risks if the models are imperfect or biased.

**Claims And Evidence:**

Yes

**Requested Changes:**

1.	Comparison to Existing Models:
o	Provide a more comprehensive comparison with existing spatiotemporal models, such as spatial-temporal attention mechanisms or condition-aware GNNs, and clarify how the proposed method improves upon them.
2.	Ablation Studies:
o	Conduct ablation studies to evaluate the individual contributions of Fourier transforms and the multi-term ensemble strategy, helping readers understand which components drive performance improvements.
3.	Handling of New Operational Conditions:
o	Offer a clearer strategy for handling unseen operational conditions during inference, potentially drawing on existing work in transfer learning or condition-specific embeddings.
4.	Broader Dataset Evaluation:
o	In addition to the C-MAPSS dataset, evaluate the model on more recent and realistic datasets, such as the dataset provided by Chao et al. (Chao, Manuel Arias, Chetan Kulkarni, Kai Goebel, and Olga Fink. "Aircraft Engine Run-to-Failure Dataset under Real Flight Conditions for Prognostics and Diagnostics." Data 6, no. 1 (2021): 5. https://doi.org/10.3390/data6010005) to ensure robustness in practical applications.

**Strengths And Weaknesses:**

Strengths:
•The paper introduces a novel use of Fourier transforms in graph neural networks to improve spatiotemporal modeling for RUL prediction.
•The multi-term ensemble approach is interesting and may provide better generalization across varying operational conditions.
•Empirical results show statistically significant performance improvements over baseline models, particularly in common benchmark datasets like C-MAPSS.
Weaknesses:
•The paper does not sufficiently compare the proposed model to existing integrated spatiotemporal models, such as attention mechanisms or dual-branch networks.
•The choice of Fourier transforms is underexplained, and there is little discussion on the potential loss of local temporal details, which could affect RUL predictions.
•The datasets used for evaluation may not fully reflect real-world operational variability, limiting the generalizability of the results.
•Some sections of the paper, especially the methodological descriptions, are difficult to follow due to complex language and notation.

---

> ### Author Response · Authors · 2024-11-08
> **Author Response (1/3)**
>
> Thank you for your critical reviews! We appreciate your insight and respond below to your concerns:
>
> **Q1: Provide a detailed comparison with existing spatiotemporal models, such as spatial-temporal attention mechanisms or condition-aware GNNs, highlighting the advantages of the proposed approach.**
>
> **A1:** In our proposed MT-FGNE framework, we introduce several innovative approaches that differentiate it from conventional spatiotemporal models like ST-GNNs. The key distinctions and advantages can be summarized in three main aspects:
>
> Novel Data Representation and Processing: Instead of following the traditional ST-GNN approach of viewing samples as sequences of graphs, we represent samples as complete graphs and transform them into the frequency domain using Discrete Fourier Transform (DFT). This transformation, combined with Fourier Graph Neural Networks (FGN), enables our model to directly capture degradation trends without the need for separate spatial and temporal modeling, thus leading to more effective learning of spatiotemporal interdependencies in sensor signal data.
>
> Multi-Term Ensemble Learning: To address the common challenge of variable-length time series in real-world applications, we introduce a multi-term ensemble learning framework. This approach specifically tackles the inadequate learning of long-term dependencies, a common limitation in existing models. Our experimental results demonstrate that this ensemble strategy significantly enhances the predictive performance of the base FGN model.
>
> Sequence Decomposition for Multiple Operation Conditions: We address the specificity of degradation under multiple operation conditions through a sequence decomposition strategy. By decomposing and interpolating the original signals, our approach effectively mitigates the performance degradation typically observed when FGNs are applied to data from multiple operation conditions. This makes our model more robust and adaptable to complex real-world scenarios.
>
> To address the reviewer's concern, we have substantially revised our model description to clearly highlight these distinguishing features and their advantages over existing approaches. Our experimental results validate the effectiveness of these innovations, particularly in handling complex spatiotemporal dependencies and multiple operating conditions.
>
> **Q2: Conduct ablation studies to identify the contributions of Fourier transforms and the ensemble strategy to performance improvements.**
>
> **A2:** Following the reviewer's suggestion, we have conducted comprehensive additional ablation studies to analyze the individual and combined effects of three key components: GNN with Fourier transforms (FGN), Multi-Term Ensemble strategy (MTE), and Sequence Decomposition (SD).
>
> The updated results in `Table 4` reveal several important findings across different datasets. For single operation condition datasets (FD001 & FD003), the combination of FGN and MTE significantly improves prediction performance, while SD is not necessary due to the inherent consistency in operation conditions. In contrast, for multiple operation conditions datasets (FD002 & FD004), both SD and MTE substantially enhance FGN's performance. Particularly interesting findings emerged when using CNN as the base models. The SD component shows limited benefits when used with CNN, which can be attributed to different learning mechanisms. While SD processing increases signal stationarity, which particularly benefits FGN's frequency domain learning, CNNs excel at capturing local temporal patterns and amplitude variations directly. Notably, the simple combination of CNN and MTE outperforms all existing state-of-the-art models on FD002 & FD004. These comprehensive ablation studies demonstrate that the effectiveness of each component varies based on the dataset characteristics, while the synergistic effect of FGN and MTE is consistently beneficial across all datasets. Additionally, SD's contribution is most significant when combined with frequency-domain learning approaches, and the choice of base model influences the effectiveness of different components. Through this analysis, we have validated the contribution of each architectural component to the overall model performance and provided insights into their interaction effects.
>
> To address the reviewer's concern, we have revised the Ablation studies section in the manuscript to clarify these points.

---

> > ### Author Response · Authors · 2024-11-08
> > **Author Response (2/3)**
> >
> > **Q3: Develop a strategy for handling unseen operational conditions during inference, possibly using transfer learning or condition-specific embeddings.**
> >
> > **A3:** Encountering unseen operational conditions during inference is indeed a common challenge in real-world applications. We propose a systematic strategy to address this challenge by leveraging the Silhouette Coefficient as a reliable metric to detect potential unseen operational conditions. Our approach is based on the observation that the presence of unseen conditions can significantly impact the quality of sequence decomposition using k-means clustering, as these new operational states may not conform to the existing cluster structure.
> >
> > We validated our strategy through extensive experiments on FD002 dataset. The results demonstrated a strong correlation between clustering quality (measured by Silhouette Coefficient, ranging from -1 to 1) and prediction performance. Specifically, with the default setting of n_cluster = 6, we achieved both the highest clustering quality (silhouette_score = 0.92) and the best prediction performance (RMSE = 13.23). In comparison, alternative settings of n_cluster = 4 (silhouette_score = 0.88, RMSE = 13.79) and n_cluster = 8 (silhouette_score = 0.80, RMSE = 14.81) showed inferior results, confirming that our chosen configuration effectively captures the underlying operational conditions structure.
> >
> > This approach provides a practical solution for real-world scenarios where the number of operational conditions is unknown. By monitoring the Silhouette Coefficient during inference, we can effectively detect the presence of unseen conditions through observed degradation in clustering quality. When such degradation is detected, it serves as an early warning signal that the model may need to be adjusted or retrained to accommodate these new conditions. We have incorporated these analysis experiments and findings into the manuscript to demonstrate the robustness and practicality of our approach.
> >
> > **Q4: Expand evaluation to more realistic datasets, including the Chao et al. aircraft engine dataset, to confirm the model's robustness.**
> >
> > **A4:** Thank you for suggesting the evaluation of more realistic datasets. Following your recommendation, we have extended our experiments to include the N-CMAPSS dataset introduced by Chao et al. [1], which provides a more comprehensive and realistic simulation of aircraft engine operations. This dataset represents a significant advancement over C-MAPSS by incorporating complete flight phases (climb, cruise, and descent) and offering more complex, real-world operational scenarios, making it an excellent benchmark for assessing our model's robustness and generalization capabilities.
> >
> > We followed the experimental setup outlined in the literature, adopting the data split methodology from Chao et al. [1] and the detailed settings from Wang et al. [2]. Specifically, we utilized six units (2, 5, 10, 16, 18, and 20) for training, while units 11, 14, and 15 served as the test set. To ensure a thorough and fair comparison with the state-of-the-art models presented by Wang et al. [2], we evaluated our model on the complete degradation trajectories of individual test units as well as on the entire test dataset. The experimental results shown in the table below demonstrate the superior performance of our proposed MT-FGNE compared to state-of-the-art methods across different units on the N-CMAPSS dataset. Across different units, our method consistently achieves lower score metrics compared to baseline models, with significant reductions of up to 40% in score values compared to transformer-based methods like Transformer and Informer. Although LOGO achieves slightly better RMSE values in some cases, MT-FGNE demonstrates superior performance in terms of the score metric, which is a more comprehensive evaluation criterion for the N-CMAPSS dataset. In summary, incorporating the N-CMAPSS dataset into our evaluation provides a thorough assessment of the model's generalization capabilities in realistic settings, demonstrating its applicability to complex, real-world degradation modeling tasks.

---

> > > ### Author Response · Authors · 2024-11-08
> > > **Author Response (3/3)**
> > >
> > > | Models | Unit11 | | Unit14 | | Unit15 | | All | |
> > > |--------|---------|---------|---------|---------|---------|---------|---------|---------|
> > > | | RMSE | Score | RMSE | Score | RMSE | Score | RMSE | Score |
> > > | Transformer [3] | 5.86 | 7725 | 7.69 | 2397 | 5.34 | 3391 | 6.54 | 17075 |
> > > | Informer [4] | 6.28 | 8019 | 7.67 | 2437 | _5.03_ | 3195 | _6.24_ | 16066 |
> > > | Autoformer [5] | 10.73 | 23987 | 13.72 | 9533 | 11.29 | 15383 | 11.48 | 50322 |
> > > | Crossformer [6] | 6.89 | 11816 | 8.58 | 2855 | 7.39 | 5733 | 6.87 | 16704 |
> > > | DAG [7] | 9.00 | 21282 | 9.37 | 4211 | 7.30 | 7970 | 8.82 | 36780 |
> > > | STGCN [8] | 8.96 | 32080 | 11.96 | 17665 | 8.39 | 20169 | 9.31 | 37710 |
> > > | HAGCN [9] | 6.39 | 9956 | _7.03_ | 2009 | 7.48 | 9376 | 6.67 | 17918 |
> > > | MAGCN [10] | 8.33 | 13835 | 10.52 | 4566 | 5.11 | 3623 | 7.37 | 20821 |
> > > | LOGO [2] | **5.73** | _7509_ | **6.72** | _1940_ | **4.54** | 3017 | **6.07** | _15127_ |
> > > | MT-FGNE | _6.16_ | **4711** | 8.12 | **1347** | 5.81 | **2388** | 6.32 | **8447** |
> > >
> > > Reference:
> > >
> > > [1] Arias Chao, M., Kulkarni, C., Goebel, K., & Fink, O. (2021). Aircraft engine run-to-failure dataset under real flight conditions for prognostics and diagnostics. Data, 6(1), 5.
> > >
> > > [2] Wang, Y., Wu, M., Jin, R., Li, X., Xie, L., & Chen, Z. (2023). Local–global correlation fusion-based graph neural network for remaining useful life prediction. IEEE Transactions on Neural Networks and Learning Systems.
> > >
> > > [3] Mo, Y., Wu, Q., Li, X., & Huang, B. (2021). Remaining useful life estimation via transformer encoder enhanced by a gated convolutional unit. Journal of Intelligent Manufacturing, 32(7), 1997-2006.
> > >
> > > [4] Zhou, H., Zhang, S., Peng, J., Zhang, S., Li, J., Xiong, H., & Zhang, W. (2021, May). Informer: Beyond efficient transformer for long sequence time-series forecasting. In Proceedings of the AAAI conference on artificial intelligence (Vol. 35, No. 12, pp. 11106-11115).
> > >
> > > [5] Wu, H., Xu, J., Wang, J., & Long, M. (2021). Autoformer: Decomposition transformers with auto-correlation for long-term series forecasting. Advances in neural information processing systems, 34, 22419-22430.
> > >
> > > [6] Zhang, Y., & Yan, J. (2023, May). Crossformer: Transformer utilizing cross-dimension dependency for multivariate time series forecasting. In The eleventh international conference on learning representations.
> > >
> > > [7] Li, J., Li, X., & He, D. (2019). A directed acyclic graph network combined with CNN and LSTM for remaining useful life prediction. IEEE Access, 7, 75464-75475.
> > >
> > > [8] Wang, M., Li, Y., Zhang, Y., & Jia, L. (2021). Spatio-temporal graph convolutional neural network for remaining useful life estimation of aircraft engines. Aerospace Systems, 4(1), 29-36.
> > >
> > > [9] Li, T., Zhao, Z., Sun, C., Yan, R., & Chen, X. (2021). Hierarchical attention graph convolutional network to fuse multi-sensor signals for remaining useful life prediction. Reliability Engineering & System Safety, 215, 107878.
> > >
> > > [10] Chen, L., Chen, D., Shang, Z., Wu, B., Zheng, C., Wen, B., & Zhang, W. (2023). Multi-scale adaptive graph neural network for multivariate time series forecasting. IEEE Transactions on Knowledge and Data Engineering, 35(10), 10748-10761.
> > >
> > > We hope that our responses have solved your problems. Thank you for your thoughtful comments again. We also appreciate your feedback regarding the paper's readability and description of the methodology. Following your suggestions, we have thoroughly revised the methodological sections to enhance clarity.

---

### Comment · Reviewer_9m7r · 2024-12-08
**The authors have carefully modified the paper.**

This paper can be accepted.

---

### Decision · Action_Editor_j2Qo · 2024-12-24

**Recommendation:** Accept with minor revision

**Comment:**

This paper proposes a practical approach towards remaining useful life prediction. The approach consists of a Fourier graph neural network, a multi-term ensemble learning framework, and a sequence decomposition method. Empirical results validate the effectiveness of the approach.

While all reviewers recommended (weak) acceptance, the writing and format of the paper needs improvement. For example, Fig 1 is not readable as the modules are not expanded, and the upright room is all wasted; Misuses in citations format (please tell between \cite, \citeyear, and \citeauthor); just to name a few. Please carefully polish the paper and submit again.

**Audience:**

Researchers working in the field of applying AI to remaining useful life prediction, especially in the areas of equipment and manufacturing, will be interested in this study.

**Claims And Evidence:**

It has been confirmed by the reviewers and the AE that the claims are well supported by sufficient evidence.

**Resubmission Of Major Revision:**

The authors may consider submitting a major revision at a later time.

---

> ### Author Response · Authors · 2025-01-16
> **Author Response**
>
> The authors thank the reviewers and the Associate Editor for their valuable feedback. In response to the detailed comments regarding the writing and formatting of the paper, we have made the following improvements.
>
> 1. **Figure 1 Optimization**: The layout has been redesigned for better visual presentation.
> 2. **Citation Style**: All citations have been reformatted according to the journal guidelines.
> 3. **Language Enhancement**: Informal expressions have been replaced with more precise academic language.
> 4. **Mathematical Notation**: Equation notation and formatting have been standardized throughout the manuscript.
>
> We have carefully addressed all the comments to improve the overall quality of our manuscript. Thank you for your valuable feedback and consideration of our revised submission.